# Thyroid hormone induces progression and invasiveness of squamous cell carcinomas by promoting a ZEB-1/E-cadherin switch

Caterina Miro[1,9], Emery Di Cicco[1,9], Raffaele Ambrosio[2], Giuseppina Mancino[1], Daniela Di Girolamo[1], Annunziata Gaetana Cicatiello[1], Serena Sagliocchi[1], Annarita Nappi[1], Maria Angela De Stefano[1], Cristina Luongo[1], Dario Antonini[3], Feliciano Visconte[4], Silvia Varricchio[5], Gennaro Ilardi [5], Luigi Del Vecchio[4], Stefania Staibano[5], Anita Boelen[6], Cedric Blanpain [7], Caterina Missero [3,4], Domenico Salvatore[4,8] & Monica Dentice[1,4]*

Epithelial tumor progression often involves epithelial-mesenchymal transition (EMT). We report that increased intracellular levels of thyroid hormone (TH) promote the EMT and malignant evolution of squamous cell carcinoma (SCC) cells. TH induces the EMT by transcriptionally up-regulating ZEB-1, mesenchymal genes and metalloproteases and suppresses E-cadherin expression. Accordingly, in human SCC, elevated D2 (the T3-producing enzyme) correlates with tumor grade and is associated with an increased risk of postsurgical relapse and shorter disease-free survival. These data provide the first in vivo demonstration that TH and its activating enzyme, D2, play an effective role not only in the EMT but also in the entire neoplastic cascade starting from tumor formation up to metastatic transformation, and supports the concept that TH is an EMT promoter. Our studies indicate that tumor progression relies on precise T3 availability, suggesting that pharmacological inactivation of D2 and TH signaling may suppress the metastatic proclivity of SCC.

[1] Department of Clinical Medicine and Surgery, University of Naples "Federico II", Naples, Italy. [2] IRCCS SDN, Naples, Italy. [3] Department of Biology, University of Naples "Federico II", Naples, Italy. [4] CEINGE–Biotecnologie Avanzate Scarl, Naples, Italy. [5] Department of Advanced Biomedical Sciences, University of Naples "Federico II", Naples, Italy. [6] Endocrine Laboratory, Department of Clinical Chemistry, Amsterdam University Medical Center, location AMC, Amsterdam, The Netherlands. [7] IRIBHM, Université Libre de Bruxelles (ULB), Brussels, Belgium. [8] Department of Public Health, University of Naples "Federico II", Naples, Italy. [9] These authors contributed equally: Caterina Miro, Emery Di Cicco. *email: monica.dentice@unina.it

Non-melanoma skin cancer (NMSC) is the most common cancer in humans and include, basal cell carcinomas (BCCs) and squamous cell carcinomas (SCCs)[1]. BCC is highly proliferative but rarely metastatic, whereas cutaneous squamous cell carcinoma (cSCC) can metastasize primarily to lymph nodes and then to the liver and lungs, and accounts for ~20% of annual skin cancer-associated mortalities[2]. Both BCC and cSCC are primarily induced by sunlight exposure and a combination of environmental, genetic, and phenotypic factors[1].

Thyroid hormones (TH) T4 and T3 are pleiotropic agents that regulate the metabolism and growth of many cells and tissues, and thus have a strong impact on cancer[3]. THs act by binding to TH receptors in the nucleus and activating or repressing target genes[3]. From development to adult life, TH signaling regulates cell-fate determination and differentiation in normal and pathological contexts[4]. According to the current paradigm of TH action, the local intracellular modulation of TH concentration is the critical determinant of cellular TH signaling[5].

Intracellular TH signaling is adapted within target cells via the action of the deiodinase enzymes D2 and D3. In detail, D2 confers on cells the capacity to increase T3 production thereby enhancing TH signaling, whereas D3 has the opposite effect[6]. D3, which is an oncofetal protein, rarely expressed in adult life, is re-activated in proliferating and neoplastic contexts. We previously demonstrated that D3 is overexpressed in BCCs and that this event is under the control of the Shh pathway[7]. Attenuation of TH by D3 in BCC cells is sufficient to enhance susceptibility to cancer formation[7]. Accordingly, TH treatment reduces tumor growth by attenuating the oncogenic potential of the BCC tumor drivers Shh and miR21[8,9]. The anti-tumorigenic action of TH in BCC can be attributed to its ability to reduce tumor cell proliferation, and increase the apoptotic rate[10,11]. However, how TH impacts on the progression, invasiveness, and metastasis of skin cancers is largely unknown.

The aim of this study was to determine the effects of TH signaling and its regulators in the late stages of the neoplastic process. To this aim, we investigated how the deiodinases are regulated during cSCC progression, and evaluated whether and how perturbation of the deiodinase-driven regulation of TH metabolism functionally impacts on the behavior of cSCC. We used the classical two-step carcinogenesis model in mouse skin, that results in the formation of benign tumors, progression to aggressive cSCC and, finally, to invasive tumors[12,13]. Gene profiling in deiodinases-depleted SCC cells by CRISPR/Cas9 technology revealed the molecular mechanisms by which T3 regulates invasiveness and migration of SCC cells. We demonstrate that the D2-mediated T3 production promotes the transcription of ZEB-1, which activates the epithelial–mesenchymal transition (EMT) cascade, and increases the invasiveness of SCC cells. Importantly, analysis of metastasis to distant sites demonstrated that D2 is a common marker of metastatic cells. Moreover, in human samples, D2 was associated with an increased risk of postsurgical relapse and with malignant grading of a large cohort of tumors.

Taken together, our in vitro and in vivo experiments revealed that the effects of D2 and D3, and consequently, TH action, are uncoupled to the various phases of cSCC tumorigenesis. Indeed, while TH initially reduces tumor formation, it subsequently accelerates invasiveness and metastatic conversion. Our data demonstrate that TH and its modulating enzymes D2 and D3 are critical microenvironmental determinants of tumorigenesis.

## Results

**TH signaling is dynamically regulated during tumorigenesis.** To investigate the role of TH and its metabolism in the progression, invasiveness, and metastasis of skin cancers, we studied the expression of D2 and D3 in different stages of SCC using the two-step chemically induced carcinogenesis model[14]. In the first, initiation step, mice were treated with the mutagen DMBA; in the second promoting step, mice were treated with 12-O-tetra-decanoyl-phorbol-13-acetate (TPA), a drug that stimulates epidermal proliferation and inflammation (Fig. 1a). Because of the lack of functional D2 antibodies in commerce, carcinogenesis experiments were performed in the knock-in D2-Flag mouse[15] in which the endogenous gene coding for D2 is fused to the Flag tag. As shown in Figs. 1b and S1, D3 rapidly increased during the initial tumorigenic step and peaked at the hyperplastic epidermis stage. D3 was highly expressed up to the formation of papillomas, which are benign intermediate lesions in the multistage progression to skin carcinoma[16] (Fig. 1b and Supplementary Fig. 1). D2 expression began at later time points and reached a nadir at the final phases of tumorigenesis when papillomas lose their differentiation potential, become more invasive and turn into SCCs (Fig. 1b, c and Supplementary Fig. 1). D2 was closely associated with the expression of K8, which is a simple epithelial keratin absent in normal epidermis but often detected in advanced SCCs[17]. mRNA analysis confirmed the sequential expression of D3 and D2 at 15 and 30 weeks after DMBA/TPA treatment (Fig. 1c). Notably, the dynamic expression of D2 and D3 is consistent with lower intra-tumoral T3 levels in the papillomas then in the more advanced SCCs (Fig. 1d). Co-expression of D2 and D3 with markers of specific stages of tumorigenesis was confirmed by western blot and immunofluorescence analysis (Fig. 1b, d and f). In detail, D2 expression was associated with reduced E-cadherin and enhanced vimentin expression, which is a sign of the EMT typical of advanced SCCs (Fig. 1f). Immunofluorescence analysis also showed that D2 was highly expressed in the late stages of tumorigenesis together with K8 and vimentin, although its expression only partially co-localized with K8 and vimentin (Supplementary Fig. 2). Thus, our results show that the EMT of SCC coincides with a switch in D3-D2 deiodinases. Indeed, while D3 is a marker of the initial stages of tumorigenesis, D2 expression is associated with cancer progression.

**Sustained TH signaling accelerates tumor invasion.** To gain insight into the role of TH in tumor progression, we disrupted the TH signal balance by depleting D2 or D3 in the epidermal compartment. To this end, we crossed D3[fl/fl18] or D2[fl/fl19] mice with K14Cre[ERT] mice[20]. First, we depleted D3 by treating K14Cre[ERT+/−], D3[fl/fl] (skin D3KO, sD3KO) and K14Cre[ERT−/−], D3[fl/fl] (CTR) mice with tamoxifen, and applied DMBA/TPA to their dorsal skin one week later (Fig. 2a). Effective D3-depletion from the epidermal compartment was confirmed by PCR and immunofluorescence (Supplementary Fig. 3A, B). Dorsal skin and skin lesions were collected 8 and 15 weeks after treatment. Eight weeks after DMBA treatment, the analysis of skin morphology and of the markers of epidermal hyperplasia, i.e., K14 and K6, showed that tumor formation was much lower in sD3KO mice than in controls (Fig. 2b). Accordingly, tumor formation was attenuated in sD3KO mice 15 weeks after DMBA treatment. D3-depletion reduced the frequency (number of tumors at each time point) and the incidence (time of lesion appearance) of cancers (Fig. 2c). Strikingly, based on morphological features, the few skin lesions observed in the sD3KO mice were more advanced lesions than those in control mice (Fig. 2d). Accordingly, K8 expression was higher in D3KO mice than in control mice, which suggests that D3-depletion accelerates SCC formation (Fig. 2e). The E-cadherin/N-cadherin ratio was lower in D3KO lesions than in control lesions, which confirms the more advanced stage of these lesions in D3KO mice (Fig. 2f, g). These results indicate that TH

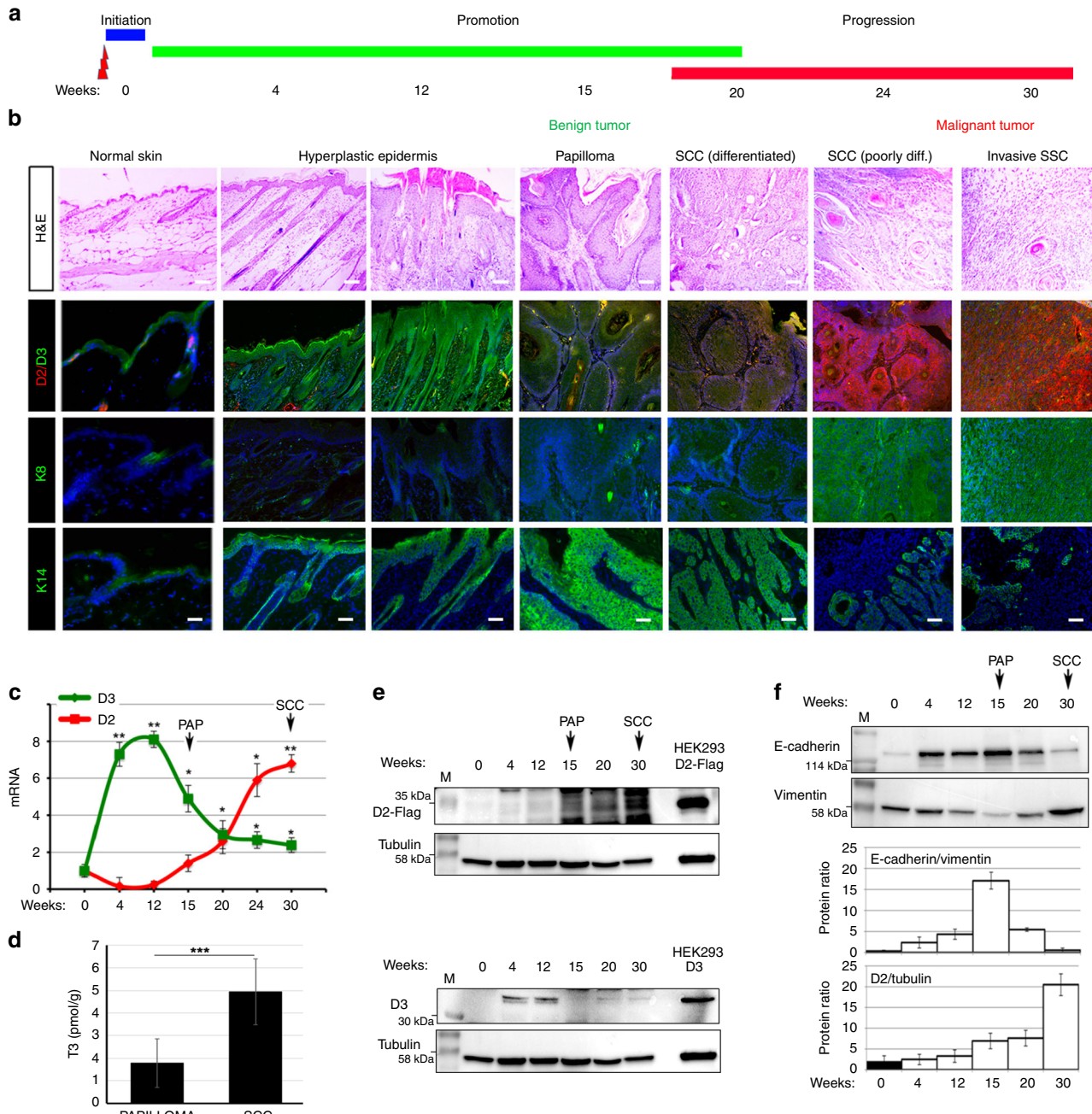

**Fig. 1** D2 and D3 are dynamically expressed during SCC tumor initiation and progression. **a** Schematic representation of the two-step carcinogenesis experiment. **b** Representative H&E, D2/D3 co-staining, and K8 and K14 staining of DMBA-TPA-treated skins for the indicated weeks. Images represent tumors of one of eight different D2-Flag mice analyzed for each time point (*n* = 8). Scale bar represents 200 µm. **c** mRNA levels of D2 and D3 were measured in progressive DMBA-TPA-treated skins as indicated in **a** and **b**, by real-time PCR analysis (*n* = 8). **d** Intracellular T3 was measured as indicated in the "Methods" section in dorsal skin tissues from mice treated with DMBA-TPA for 15 (PAP) and 30 (SCC). **e** Western blot analysis of D2-Flag and D3 expression from skin lesions of D2-3xFlag mice treated with DMBA-TPA for the indicated times (*n* = 8). HEK293 cells transfected with D2-Flag or D3 served as D2 and D3 controls, respectively. Arrows indicate the papillomas (PAP) and SCC tumors. **f** Western blot analysis of E-cadherin, vimentin, and tubulin in the same samples as in **e**. Quantification of the single protein levels versus tubulin levels and the E-cadherin/vimentin ratio is represented by histograms.

plays an unexpected stage-specific role in SCC development, i.e., T3 slows the early stages of tumorigenesis, but induces SCC progression (Fig. 2h).

**Attenuation of TH signaling reduces tumor invasion.** To evaluate tumor progression in an epidermal-specific D2KO background (sD2KO), we depleted D2 from the epidermal compartment by administrating tamoxifen before the DMBA/TPA treatment and analyzed the skin lesions 20 weeks later (Fig. 3a). As shown in Fig. 3b, c, D2-depletion increased the frequency of hyperplastic lesions and papillomas. sD2KO mice developed skin papillomas at an average of 10 weeks after the first application of DMBA, compared with 14 weeks in control mice (Fig. 3b). By week 20, 100% of sD2KO mice and only 70% of control mice developed tumors. D2-depletion increased the level

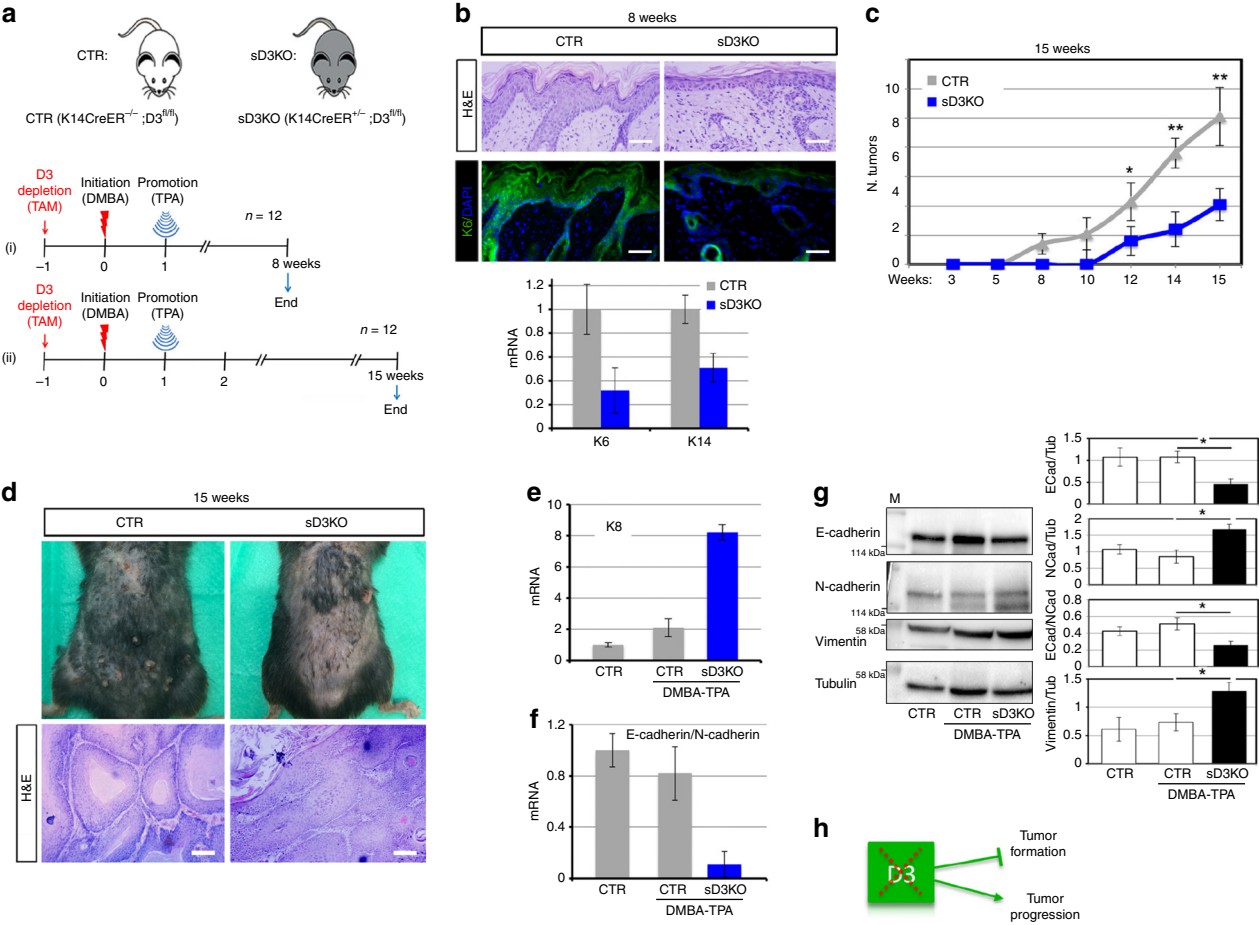

**Fig. 2** Increased thyroid hormone action in epidermal-specific D3KO mice enhanced the migration and invasion potential of SCC. **a** Schematic representation of D3-depletion and the two-step carcinogenesis experiment in 12 CTR and 12 sD3KO samples ($n = 12$). **b** Representative H&E and K6 staining of DMBA-TPA-treated CTR ($n = 8$) and sD3KO ($n = 8$) skins for 8 weeks. Scale bar represents 200 μm (top). mRNA levels of K6 and K14 were measured by real-time PCR analysis in 16 lesions from CTR mice and 16 lesions from sD3KO mice (bottom). **c** The number of skin lesions was counted during the two-step carcinogenesis experiment as indicated in **a** (ii). Tumor incidence is expressed as the number of tumors per mouse. **d** Representative images of the dorsal skin from CTR and sD3KO mice treated with DMBA-TPA for 15 weeks ($n = 12$). H&E of the skin lesions from CTR ($n = 6$) and sD3KO ($n = 6$) mice treated with DMBA-TPA for 15 weeks. Scale bar represents 200 μm. **e** mRNA levels of K8 in the dorsal skin of CTR ($n = 15$) and sD3KO ($n = 15$) mice. **f** mRNA levels of E-cadherin/N-cadherin in skin lesions of CTR ($n = 15$) and sD3KO ($n = 15$) mice measured by real-time PCR analysis. **g** Western blot analysis of E-cadherin, N-cadherin, and vimentin expression in skin lesions of CTR ($n = 10$) and sD3KO mice ($n = 10$). Quantification of the single protein levels versus tubulin levels and the E-cadherin/N-cadherin ratio is represented by histograms. *$P < 0.05$, **$P < 0.01$. **h** Schematic representation of the effects of D3-depletion on SCC tumor growth and progression.

of K6 in D2KO papillomas, which confirms that lowering the level of T3 by D2-depletion accelerates tumor growth (Fig. 3d, e). Notably, the greater tumor growth in D2KO mice was not associated with greater tumor progression and invasive conversion. Indeed, as shown in Fig. 3d, e, K8 expression was lower in sD2KO mice than in control mice, which indicates that D2KO papillomas resist progression to SCC. The lower propensity of sD2KO tumors to acquire a more invasive phenotype was confirmed by the higher E-cadherin/N-cadherin ratio in sD2KO tumors (Fig. 3d, e). This finding demonstrates that TH attenuation via D2-depletion reduces the EMT, and that, while increasing tumor formation, it slows tumor progression.

To assess the impact of D2-depletion at later stages of tumor progression, we removed D2 from the epidermal compartment 20 weeks after DMBA treatment and analyzed tumor behavior thereafter (Fig. 3f). In this case, the total number of tumors was similar in the sD2KO and control mice. However, while tumors in control mice evolved towards a carcinoma-like phenotype (Fig. 3f), tumor progression was delayed in the sD2KO mice. Indeed, by week 25, sD2KO mice developed an average of 1.1

carcinoma-like, large tumors (>3 mm) per mouse compared with 4.4 large tumors per mouse in their control littermates ($P < 0.01$; Fig. 3g, h). Moreover, K8 levels were lower in sD2KO mice than in control mice (Fig. 3i), which suggests that D2 expression and TH activation is an essential component of the shift from a papilloma grade to a more invasive SCC (Fig. 3j).

**D2 and D3 are dynamically regulated in secondary tumors.** To verify whether D2 and D3 expressions actually changes during tumor progression, we used an alternative model of SCC tumorigenesis, namely transplantation of tumor-initiating cells (TICs) into immunodeficient-athymic mice. We isolated TICs by FACS sorting of Lin−/α-6integrin+/Epcam+/CD34+ cells from SCC tumors (Fig. 4a). These cells form secondary tumors upon transplantation[20]. Ten weeks after transplantation, mice developed palpable tumors (Type I tumors), which progressed and assumed a SCC-like phenotype 16 weeks after transplantation (Type II tumors) (Fig. 4b). Lymph node morphology differed between type I and type II tumor-bearing mice (Fig. 4b).

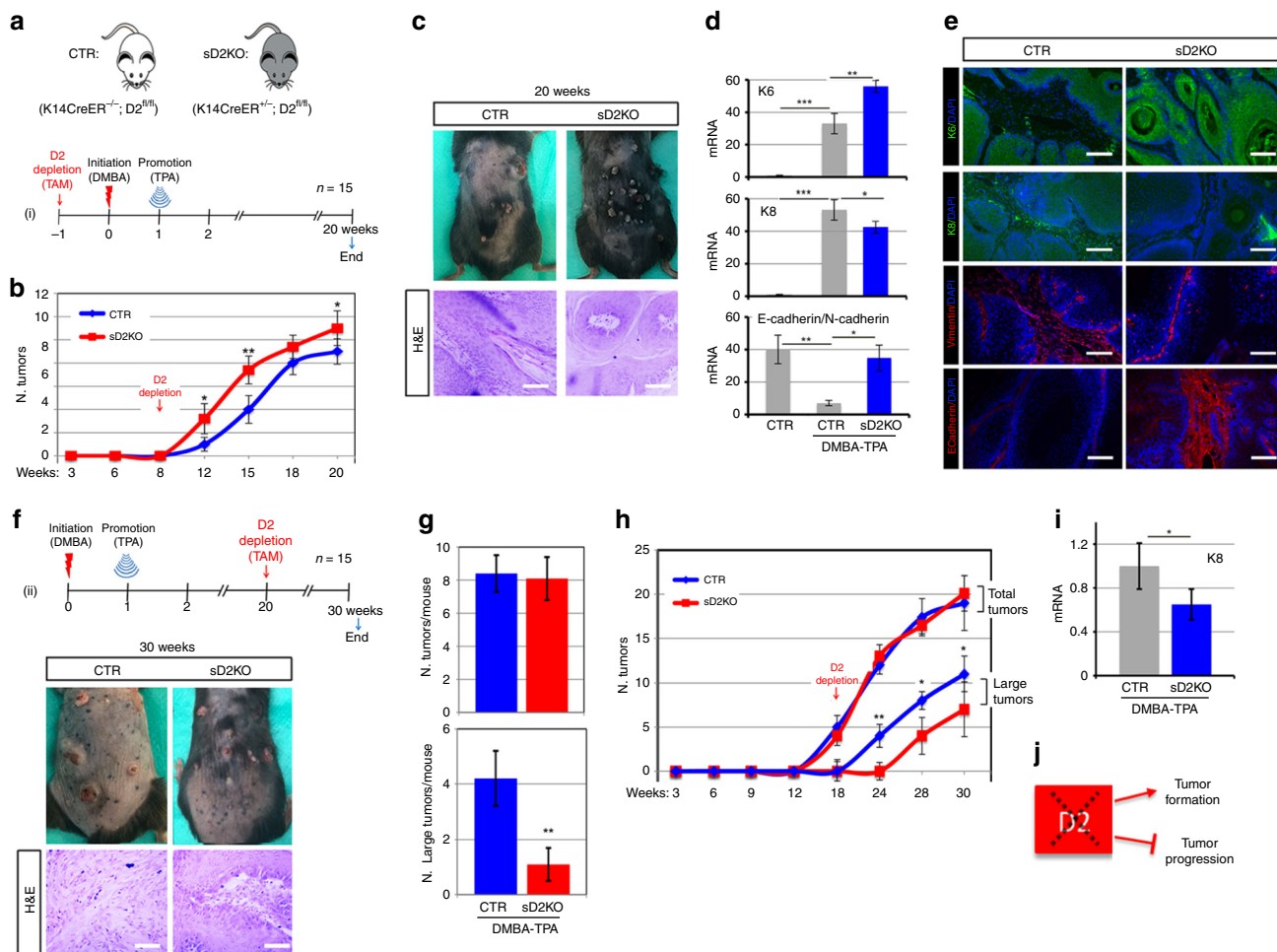

**Fig. 3** Attenuation of TH signaling in D2KO mice enhanced tumor growth and reduced the EMT. **a** Schematic representation of D2-depletion and the two-step carcinogenesis experiment in 15 CTR and 15 sD2KO samples ($n = 15$). **b** The number of skin lesions counted during DMBA/TPA treatment in sD2KO and CTR mice. **c** Picture of the dorsal skin from CTR ($n = 15$) and sD2KO ($n = 15$) mice treated with DMBA/TPA for 20 weeks showing papillomas and advanced SCC (top). H&E of the skin lesions from CTR ($n = 8$) and sD2KO ($n = 12$) mice (bottom). Scale bars represent 200 μm. **d** The mRNA levels of K6, K8, and E-cadherin/N-cadherin ratio in skin lesions of CTR ($n = 15$) and sD2KO ($n = 15$) mice measured by real-time PCR analysis. **e** Immunostaining for K6, K8, vimentin, and E-cadherin was performed on paraffin-embedded sections of dorsal skin lesions ($n = 10$ for both groups). Scale bars represent 200 μm. **f** Schematic representation of D2-depletion from the epidermal compartment at the end of DMBA/TPA treatment (top, $n = 15$). Representative pictures and H&E staining of skin lesions from sD2KO ($n = 12$) and CTR ($n = 8$) mice treated with DMBA/TPA for 30 weeks (bottom). Scale bars represent 200 μm. **g** and **h** The number of tumors (detectable tumors, >1 mm) and number of large tumors (>3 mm) counted during DMBA/TPA treatment in sD2KO and CTR mice. **i** mRNA levels of K8 in skin lesions of CTR ($n = 15$) and sD2KO ($n = 15$) mice measured by real-time PCR analysis. *$P < 0.05$, **$P < 0.01$, ***$P < 0.001$. **j** Schematic representation of the effects of D2-depletion on SCC tumor growth and progression.

Interestingly, while the expression of D2 and D3 increased to a similar extent in the transplanted TIC cells, their expression conversely fluctuated as the tumors developed. In detail, D3 expression was enhanced in type I tumors, whereas D2 was enhanced and predominated in type II tumors and in lymph nodes from mice bearing type II tumors (Fig. 4c). These data are in line with the concept that D3 is up-regulated during tumor growth thereby attenuating TH signal in benign tumors (papillomas), while D2 expression is upregulated in more aggressive and infiltrating cancer cells thereby enabling TH signal amplification at the EMT and progression to a higher tumor grade.

**Intracellular T3 induces EMT.** Given the effects of D2-depletion and D3-depletion in SCC progression and invasiveness, we investigated whether the TH signal is a positive regulator of the invasive conversion of cancer cells. Consequently, we evaluated the contribution of the TH signal to the promotion of the EMT in

the human skin squamous cell carcinoma SCC 13 cells. As shown in Fig. 5a, T3 treatment increased N-cadherin expression and slightly reduced E-cadherin expression, thereby resulting in a net attenuation of the E-cadherin/N-cadherin ratio, which indicates an increased EMT profile. We also measured the expression of two other EMT markers namely, vimentin and Twist, and found that both were positively regulated by T3 (Supplementary Fig. 4A). These results were confirmed in two other SCC cell lines (the head and neck SCC cells SCC 011 and SCC 022). In fact, T3 treatment reduced the E-cadherin/N-cadherin mRNA ratio and increased the mRNA expression of vimentin and Twist (Supplementary Fig. 5).

To explore the role of the intracellular control of TH action in EMT, we suppressed D2 or D3 expression in SCC 13 cells using CRISPR/Cas9 technology. D2-depletion and D3-depletion were verified by gene sequencing and PCR analysis, respectively (Supplementary Fig. 6). Western blot revealed that D3-depletion (which increases intracellular T3) phenocopied the

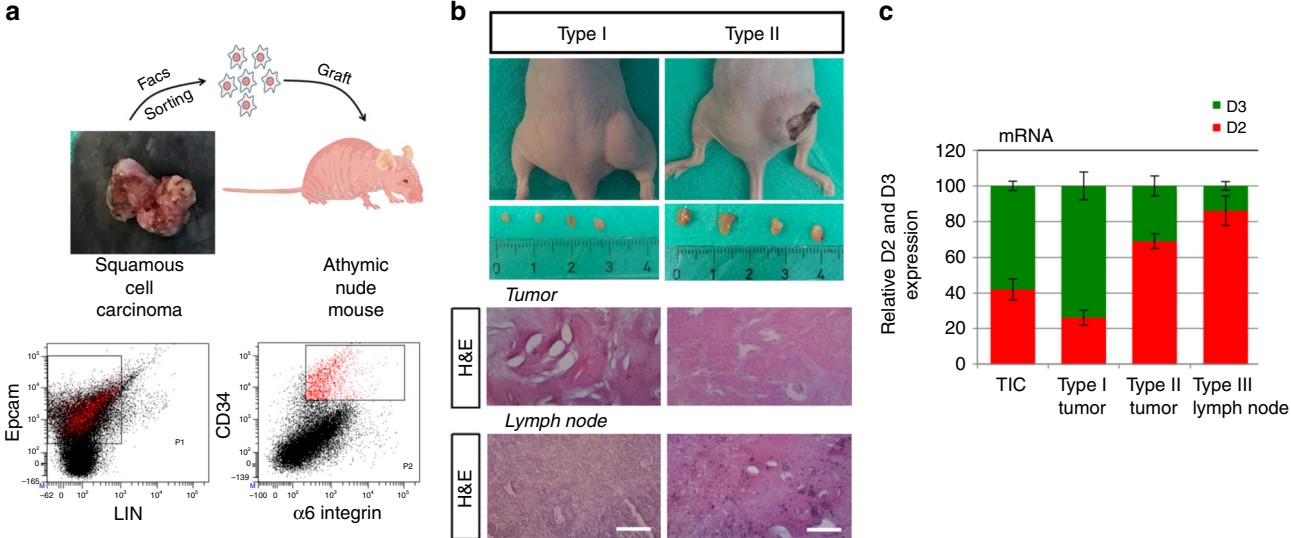

**Fig. 4** Tumor xenografts from tumor-initiating cells transplantation dynamically express D2 and D3 during tumor progression. **a** Schematic representation of the strategy used to isolate and graft TICs from DMBA/TPA tumors (*n* = 6). **b** Representative images of tumors and lymph nodes generated in athymic mice 10 (type I) and 16 (type II) weeks after transplantation (Top). H&E of the tumors and lymph nodes in type I and II mice. Scale bars represent 100 μm (Bottom). **c** Relative expression of D2 and D3 in type I and II tumors and in lymph nodes measured by real-time PCR.

effects exerted by T3 on the EMT. Indeed, the E-cadherin/N-cadherin ratio at mRNA and protein level was reduced in D3KO cells and in T3-treated cells (Fig. 5b and Supplementary Fig. 4C). Accordingly, the E-cadherin/N-cadherin ratio was elevated in D2KO cells versus control cells (Fig. 5c and Supplementary Fig. 4C). The expression of the EMT markers vimentin and Twist was also inversely regulated in D2KO cells versus D3KO cells (Supplementary Fig. 4B). Vice versa, vimentin and Twist were induced in D2 overexpressing SCC cells and down-regulated in D3 overexpressing SCC cells (Supplementary Fig. 7). D3-depletion and T3 treatment of SCC cells caused a morphological shift, namely, delocalization of phalloidin from the membrane to the cytoskeleton, an increase of vimentin expression and a decrease of E-cadherin expression (Fig. 5d, e).

Notably, T3 reduced SCC cells proliferation (Fig. 5f), which confirms its effects as an anti-proliferative agent in SCC, but contextually, it enhanced the migration of SCC cells (Fig. 5g). Accordingly, D3-depletion in D3KO cells caused growth attenuation, but enhanced migration (Supplementary Fig. 8A, C), whereas D2KO cells had the opposite profile, i.e., increased proliferation (Supplementary Fig. 8B) and reduced migration (Supplementary Fig. 8C).

To assess whether T3 induced invasiveness as well as migration, we performed a transwell invasion assay and found that T3 significantly increased the invasiveness of SCC cells (Fig. 5h), and, similarly, D3KO cells invaded the matrix more efficiently than did control cells (Supplementary Fig. 9). Since T3 induced matrix degradation (Fig. 5h), we extended the analysis to metalloproteases, which are important inducers of invasiveness and of the EMT. We analyzed the expression patterns of a panel of seven metalloproteases (MMP 2, 3, 7, 8, 9, 10, and 13) in D3KO and D2KO cells versus control cells. Real-time PCR analysis revealed that the expression of MMP 2, 3, 7, and 13 was higher in D3KO cells, and lower in D2KO cells than in controls (Supplementary Fig. 10A). Accordingly, secretion in the culture medium and the enzymatic activity of metalloproteases 2, 3, 7, and 13 were robustly increased in D3KO and reduced in D2KO cells (Supplementary Fig. 10B and C). These results show that TH acts as an up-stream regulator of cell matrix degradation and of the cell invasion process.

**Evidence of a T3-ZEB-1 functional network**. We next investigated the molecular mechanisms by which T3 induces the EMT and invasiveness of SCC cells. We profiled the expression of 84 genes whose expression is related to the EMT using the human EMT RT² Profiler™ PCR Array (Qiagen). Interestingly, most of the EMT genes were modified by D3-depletion (Fig. 6a). In particular, 22 EMT-positive regulators, among which vimentin, N-cadherin, ZEB-1, and Col1a2, were up-regulated in D3KO cells, while 13 genes were down-regulated, among which, the epidermal markers keratin 14 and E-cadherin, which indicates massive induction of the EMT-mediating cascade (Fig. 6b, Supplementary Fig. 11A and, Supplementary Data File 1). We validated the expression profile of these genes in control and D3KO cells by real-time PCR (Supplementary Fig. 11B). To identify the EMT genes that mediate TH-dependent invasiveness, we first assessed the possibility that ZEB-1, a prime transcriptional factor in the EMT cascade[21], and an up-stream regulator of many EMT-related genes[22], could be a direct target of TH. Interestingly, in silico analysis of the ZEB-1 promoter region (comprising the 5′-flanking region, 2 kb-up and 2 kb-down the TSS, Supplementary Fig. 12) revealed the presence of several thyroid hormone responsive elements (TREs)-binding sites. Chromatin IP (ChIP) confirmed that the thyroid receptor (TRα) physically binds the identified TRE regions (Fig. 6c). To assess whether ZEB-1 is a molecular determinant of the T3-dependent EMT, we down-regulated ZEB-1 in SCC cells (Supplementary Fig. 13). Importantly, a scratch assay in SCC cells indicated that ZEB-1 down-regulation (obtained using two different shRNA for ZEB-1 silencing, shZEB-1) completely rescued T3-dependent enhanced migration (Fig. 6d, e and Supplementary Fig. 14). PCR and Western blot analysis of the EMT genes confirmed that ZEB-1 depletion blocks the T3-dependent EMT cascade (Fig. 6f, g and Supplementary Fig. 14). The in vitro and in vivo relevance of the T3-ZEB-1-E cadherin axis was confirmed by measuring ZEB-1 and E-cadherin expression in SCC cells treated with T3-depleted or D3-depleted (Fig. 6h, i) and in in vivo tumors from DMBA/TPA-treated D2KO and D3KO mice. According to our model, E-cadherin expression was high and ZEB-1 expression was low in sD2KO lesions, whereas E-cadherin expression was low and ZEB-1 expression was high in sD3KO mice compared to control mice (Fig. 6j).

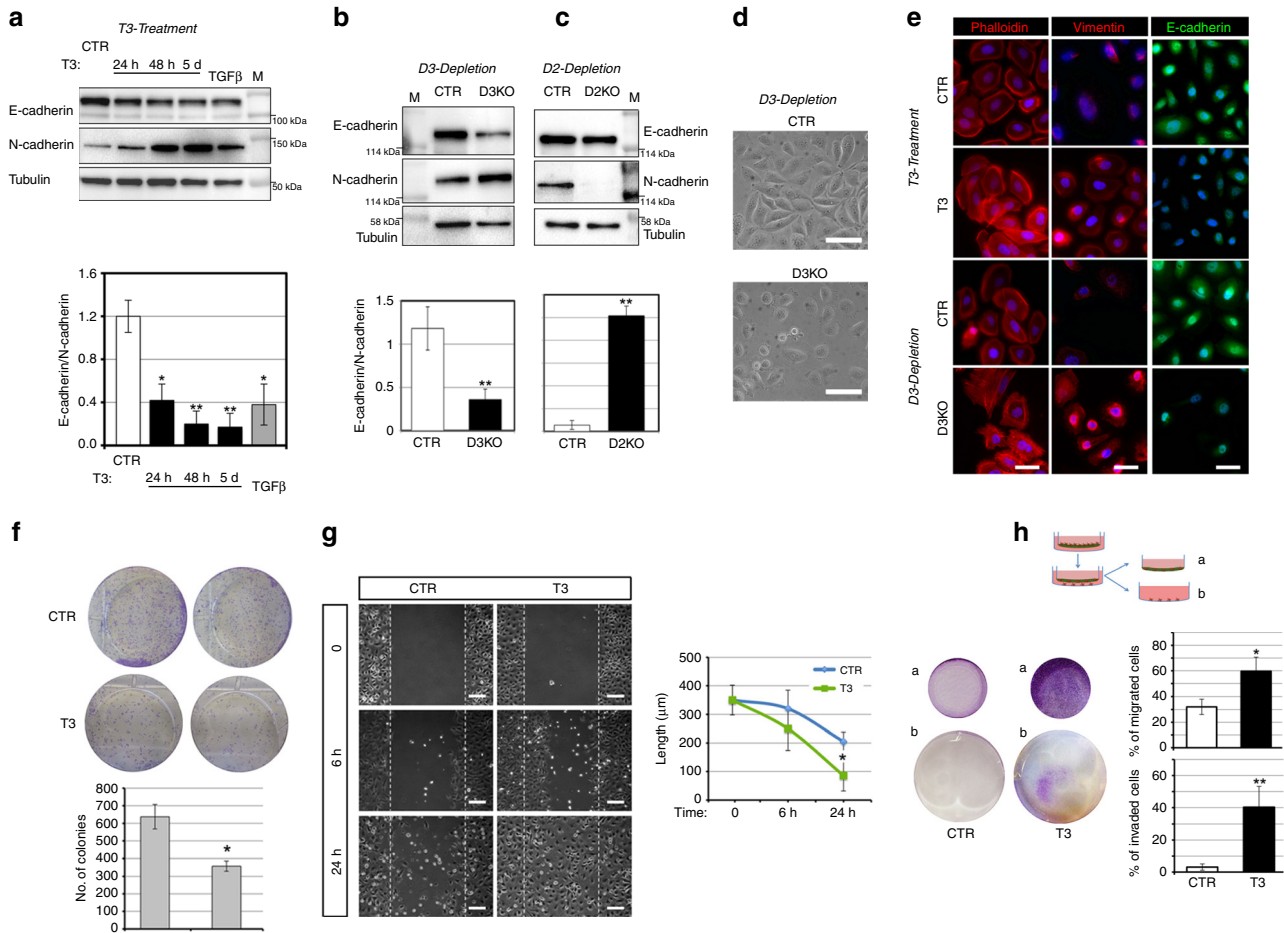

**Fig. 5** Thyroid hormone activation induces the epithelial–mesenchymal transition and migration and invasion ability of SCC cells. **a** SCC cells were treated with 30 nM T3 for different time points or with 5 ng/μl TGFβ for 48 h. Total protein lysates were used for western blot analysis of E-cadherin and N-cadherin expressions. Tubulin expression was measured as loading control. One representative immunoblot of seven is shown (top). Quantification of the E-cadherin/N-cadherin ratio in the western blot is represented by histograms. **b** Western blot analysis of E-cadherin and N-cadherin expressions performed as indicated in **a** in SCC-CTR and SCC-D3KO cells. **c** Western blot analysis of E-cadherin and N-cadherin expressions in SCC-CTR and SCC-D2KO cells. **d** Representative phase contrast of CTR and D3KO cells. Scale bars represent 50 μm. **e** Phalloidin (red), Vimentin (red), and E-cadherin staining (green) of untreated and T3-treated SCC cells, SCC-CTR cells, and SCC-D3KO cells. One representative experiment of 5 is shown. Scale bars represent 50 μm. *$P < 0.05$, **$P < 0.01$. **f** Cell proliferation was assessed by colony assay of SCC cells treated or not with 30 nM T3 (top). The number of colonies formed 8 days after plating shown by scale bars (Bottom). One representative assay of five is shown. **g** Wound scratch assay was performed in SCC cells after treatment with T3 (30 nM) for 0, 6, and 24 h. Cell migration was measured as described in the "Methods" section. Scale bars represent 200 μm. **h** Invasion assay performed on SCC cells treated or not with 30 nM T3 for 5 days. (b) Represents cells that invaded on the receiver plate and the area covered by invaded cells is indicated. The percentage of cells that migrated and invaded are represented by histograms.

**D2 depletion attenuates metastatic formation**. To study the role of T3 in metastatic formation, we first evaluated whether D2 is expressed in metastatic lesions at distant sites. Metastatic lesions in the lung of D2-3xFlag mice treated with DMBA/TPA were positive for D2, as demonstrated by double staining of D2/CXCR-4 and D2/K8 (Fig. 7a). D2 and K8 mRNA were potently expressed in lymph nodes and lung metastasis, while D3 was barely detectable in metastatic lesions (Fig. 7b). Moreover, the number of metastatic lesions in liver, lung, and lymph nodes in D2KO, D3KO, and control mice treated with DMBA/TPA for 20 and 30 weeks was higher in D3KO mice and lower in D2KO mice versus control mice (Fig. 7c).

**D2 is a prognostic marker of human SCC progression**. Since our general hypothesis is that transformation of benign papillomas to invasive SCC is causally linked to D2-mediated T3 activation, we evaluated the correlation of D2 expression in human tumors using the markers of EMT E-cadherin and ZEB-1, and the relative switch from papillomas to SCC. We collected 72 samples of human tumors at different pathologic states and of diverse tumor grade up to cSCC, to assess the clinical significance of D2 and the TH signal in human SCCs. The tumor stage of each tumor was evaluated by assessing the expression of E-cadherin and ZEB-1 by immunoistochemical assay ($n = 20$) in formalin-fixed paraffin-embedded biopsies of the five stages of the malignant evolution of keratinocytes towards cSCC: normal epidermis, actinic elastosis, advanced actinic keratosis, well-differentiated cSCC, and poorly differentiated SCC (Fig. 7d). As shown in Supplementary Data File 2, we defined a panel of four stages (T1–T4). We then measured the D2 mRNA expression by real-time PCR from the 20 biopsies and found that D2 mRNA was potently overexpressed in cSCC compared to normal epidermis, and that the highest fold change occurred in poorly differentiated cSCC (Fig. 7e). Importantly, D3 was highly expressed at the early stages of human tumorigenesis, with a peak at G1, consistent with its expression in the early stage of mouse tumor progression

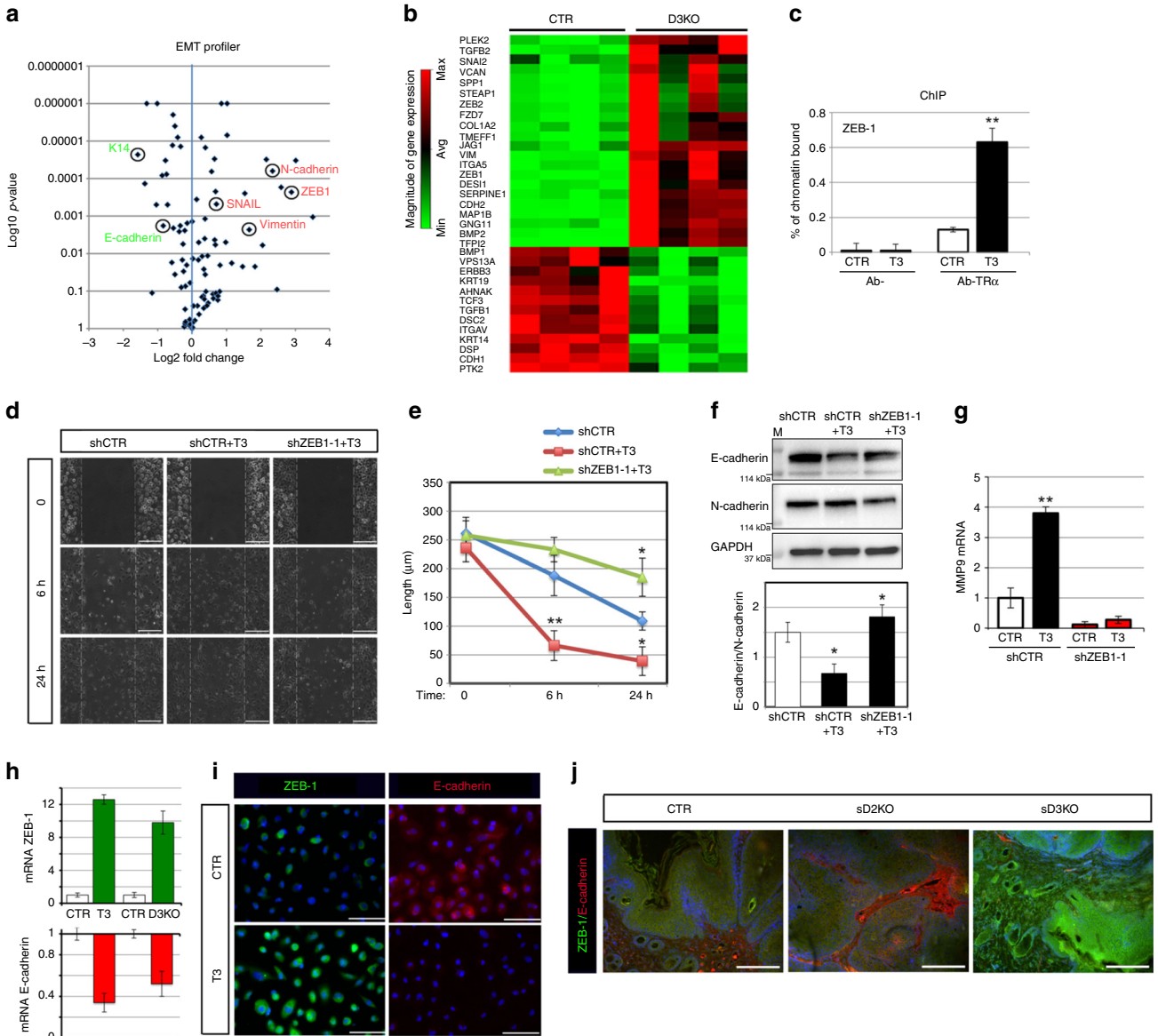

**Fig. 6** T3 promotes the EMT of SCC by inducing ZEB-1 transcription. **a** Vulcano plot of differentially expressed EMT genes measured by real-time PCR in the EMT RT² Profiler™ PCR array (Qiagen). **b** Heat map of gene expression of EMT markers in normal SCC cells versus D3-depleted SCC cells selected by thresholds of $p$ value < 0.05 and log 2 fold changes >1.3 between visits. **c** Chromatin immunoprecipitation (ChIP) of TR binding to the ZEB-1 promoter (region amplified by PCR is indicated by red arrows in Fig. S12). **d**, **e** Wound scratch assay of SCC cells after treatment with 30 nM T3 or T3 + ZEB-1 shRNA. Scale bars represent 100 μm. Scale bars represent 200 μm. **f** Western blot analysis and the relative quantification of E-cadherin and N-cadherin expressions in SCC cells treated or not with 30 nM T3 and transfected with shCTR or shZEB-1 as indicated. **g** mRNA levels of MMP9 in SCC cells treated or not with 30 nM T3 and transfected with shZEB-1 or shCTR as indicated. **h** mRNA levels of ZEB-1 and E-cadherin in T3-treated versus non-treated cells and in D3KO versus CTR SCC cells. **i** ZEB-1 (green) and E-cadherin staining (red) of untreated and T3-treated SCC cells. One representative experiment of 4 is shown. Scale bars represent 50 μm. **j** ZEB-1 (green) and E-cadherin staining (red) of skin lesions from sD2KO, sD3KO, and CTR mice. One representative experiment of 4 is shown. Scale bars represent 200 μm. *$P < 0.05$, **$P < 0.01$.

(Fig. 1b) and with the low T3 levels in papillomas (Fig. 1d). Using the X-Tile program[23], we also analyzed previous data from two collections of tumors in which the gene expression signature of SCC was associated to the risk of relapse, recurrence-free survival and overall survival of patients[24,25]. Kaplan–Meier plots from both data sets showed a striking significant correlation between high D2 levels and risk of relapse, and an inverse correlation with the percent survival of patients (Fig. 7f). As shown in Fig. 7f, high D3 levels are associated with a lower percent of survival and are not significantly associated with risk of relapse.

Taken together, the above results suggest that D2 levels are correlated with a more advanced tumor stage and with a poorer prognosis of human cancers, which suggests that the identified D2–T3–ZEB1 axis is critical in triggering invasiveness and metastatic spreading of SCCs (Fig. 7g). Should this concept be confirmed in other tumor series, it is feasible that distant-site metastasis status can be predicted from D2 gene expression levels, and that D2 might represent a marker in SCC grading procedures.

## Discussion

Tumor plasticity is a key variable in the response to therapeutic intervention. Although the classical view of the natural history of

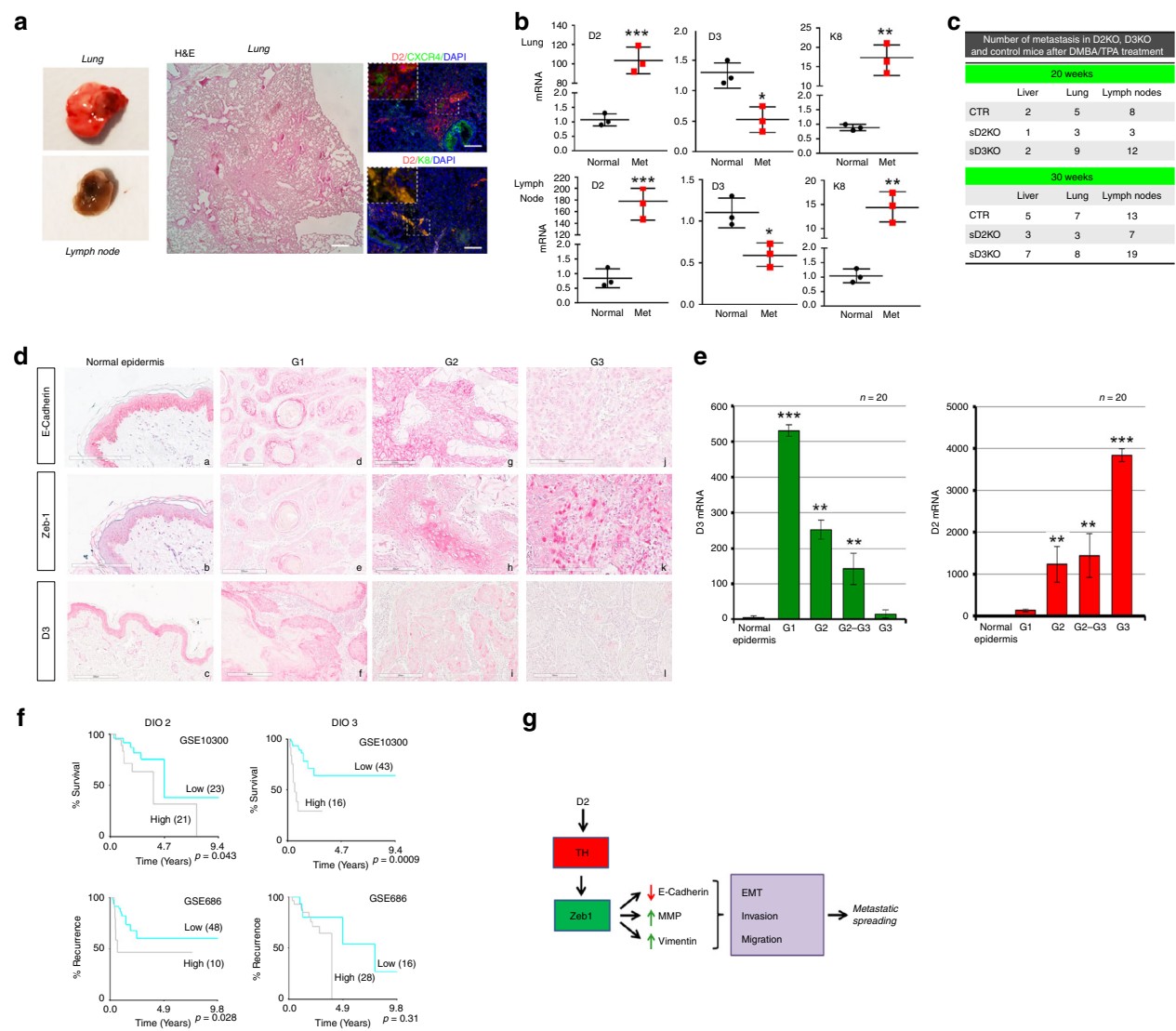

**Fig. 7** D2 is expressed in metastatic formations from SCC tumors and its expression in human tumors correlates with a poor diagnosis. **a** Representative images of metastatic lungs and lymph nodes, and H&E, D2/CXCR-4, and D2/K8 co-staining of lung sections from D2-3xFlag mice treated with DMBA/TPA for 24 weeks. Scale bars represent 200 μm. **b** Real-time PCR analysis of D2, D3, and K8 expressions in metastatic lungs and lymph nodes from DMBA-TPA-treated mice for 30 weeks (Met) and control, untreated mice (Normal). **c** Number of metastatic lesions in sD2KO, sD3KO, and control mice 20 and 30 weeks after DMBA/TPA treatment ($n = 25$). **d** Representative images showing immunohistochemical (IHC) staining for ZEB-1, E-cadherin, and D3 proteins in cutaneous squamous cell carcinoma (cSCC). Positive E-cadherin expression (a) and negative ZEB-1 expression (b) in normal tissue, and in G1 cSCC (d, e). Moderate E-cadherin expression (g) and weak ZEB-1 expression (h) in G2 cSCC. Finally, negative E-cadherin expression (j) and positive ZEB-1 expression (k) in G3 cSCC. IHC staining of D3 expression revealed positive D3 expression in normal epidermis (c), a rise in D3 expression in G1 (f) and a drastic decrease of D3 expression at the later stages G2 and G3 (i, l). IHC analysis was performed in 20 different tissues for each grading. Tumor grading was evaluated as reported in Supplementary Data File 2. **e** D3 and D2 mRNA levels were measured in human SCC samples at the same grading as in **c**. *$P < 0.05$, **$P < 0.01$, ***$P < 0.001$. **f** Kaplan–Meier plots from two independent data sets (GSE10300 and GSE 686). Blue indicates low and gray indicates high D2 or D3 expression. The number of tumors in each group is reported in parentheses. **g** The D2–T3–ZEB-1 axis contributes to the EMT and to the progression to invasive stages of carcinomas.

cancer is that tumor progression is a linear process and that metastases appear in the late stages of carcinogenesis, many clinical studies show that invasiveness can be acquired at an early stage, and result in metastasis from early neoplasms[26,27]. In this study, we demonstrate that TH activation enhances the invasiveness and metastatic propensity of SCC cells in an in vivo model of chemical cancerogenesis. Our data show that low-TH tumors (D2KO) are fast growing tumors with a low metastatic propensity. Conversely, high-TH tumors (D3KO) grow slowly but metastasize rapidly. These conclusions are based on the following results: (1) TH treatment of SCC cells reduced cell proliferation whereas it increased cell migration and invasive capability; (2) TH treatment of SCC cells also attenuated tumor formation, but accelerated the metastatic conversion of SCC; and lastly, (3) TH-attenuation in vivo accelerated tumor formation and reduced invasiveness and the development of metastatic tumors compared with wild-type littermates. These effects were mediated by a cell-autonomous mechanism of TH activation/inactivation catalyzed by the TH-modulating enzymes, D2 and D3, that turn TH on and off in target cells. During chemically induced tumorigenesis, sequential expression of D3 and D2 fine-tunes TH signaling to enable tumor growth during the early

stages of tumorigenesis, and invasive transformation at later stages (Fig. 1).

Over recent years, our understanding of TH action in cancer has evolved from being a repressor of tumor formation and growth[7,19,28] to a factor that induces most of the hallmarks that cancer cells acquire during the later stages of tumorigenesis. Indeed, our in vitro and in vivo observations demonstrate that while T3 reduced the proliferation of SCC cells, T3 treatment or D3-depletion accelerated cell migration and invasion. On the contrary, cell migration and invasion were reduced in D2KO mice. Our finding that D3-depletion reduced SCC formation (Fig. 2) confirms our previous report that the oncofetal protein D3 is an essential component of the oncogenic route leading to tumor formation. Although D3 is not a canonical oncogene per se and D3 overexpression is not sufficient for tumor initiation (unpublished data), D3-depletion drastically attenuates the growth of a variety of tumors[7,9,29]. Furthermore, D3 is positively regulated by the oncogenic pathways Shh[7], miR21[9], and Wnt[29]. Our data show that, in SCC in vivo, D3 is potently expressed in the early phases of tumorigenesis and its expression coincides with the expression of keratin K6, which is a marker of hyperplastic epidermis (Fig. 1). D3 expression declines as tumorigenesis proceeds to more invasive phenotypes. This finding is in line with our previous report that in human colon cancer, D3, which is up-regulated in carcinomas, inversely correlates with the histologic grade of lesions, and decreases as the histologic grade progresses from G2 to G3[29].

D2 has frequently been associated with cell differentiation in various cellular contexts[4,30]. Surprisingly, we recently found abundant D2 expression in BCC and SCC cells—a finding that implicated D2 in epithelial cancer[10]. A search in public databases showed that D2 expression correlates with advanced tumor grading, postsurgical relapse, and a shorter disease-free survival (Fig. 7e). Moreover, the analysis of a panel of 72 human SCC tumors revealed that D2 expression in human tumors is closely associated with more severe and invasive tumors, and that its expression closely correlates with the level of expression of ZEB-1 and negatively correlates with E-cadherin expression (Fig. 7). These data add clinical significance to our study. Indeed, they suggest that D2 expression correlates with such features of malignant tumors as short overall survival, poor prognosis, and dismal outcome. These observations reinforce the hypothesis that activation of TH is likely to be a centrally important mechanism for progression of carcinomas to a metastatic stage.

How can D2-produced T3 promote invasion? For a transformed cell to metastasize to a distant site in the body, it must first lose adhesion, and then penetrate and invade the surrounding extracellular matrix (ECM), enter the vascular system, and adhere to distant organs[31]. We found that TH modulates each of these key steps, and thus facilitates metastatic invasion of secondary sites. Indeed, the increased tumor metastatic conversion induced by T3 was not due to increased tumor cell proliferation but was correlated with (i) increased expression of mesenchymal markers, (ii) condensation of the ECM by MMP proteins, and (iii) modification of the shape and stiffness of cells. To identify the TH-dependent signature responsible for EMT induction, we used the human EMT RT² Profiler™ PCR array. The TH-positively regulated genes found included the mesenchymal markers vimentin, ZEB-1, Snail, and Col1a2. The negatively regulated genes included the epithelial-specific adhesion molecules K14 and E-cadherin. Chromatin immunoprecipitation assays revealed that among various EMT-related genes, ZEB-1 is physically bound by T3 in SCC cells and that inhibition of ZEB-1 drastically reduced the TH-dependent EMT (Fig. 6). This finding indicates that ZEB-1 is a molecular determinant that lies downstream of TH, and hierarchically mediates the TH-dependent

EMT cascade. The E-box-binding transcription factor ZEB-1 is the master regulator of the EMT that acts by enhancing the migratory and invasive capacity of cancer cells by down-regulating epithelial markers[21], thereby promoting invasion and metastasis[22]. At transcriptional level, E-cadherin is the best-known negative target of ZEB-1[32]. Moreover, ZEB-1 activates mesenchymal and stemness markers[21] thereby promoting tumorigenesis and metastasis in various human cancers[33–35]. As a repressor of E-cadherin, ZEB-1 is highly expressed in invasive carcinomas, at the invasive front of tumors in dedifferentiated cancer cells[34]. Here, we report that ZEB-1 is a down-stream target of TH. This is in line with the finding that D2-mediated TH activation is a critical step towards invasiveness and metastatic conversion of SCC. Accordingly, D2KO tumors express very low levels of ZEB-1 and high levels of E-cadherin, and they slow progression towards malignancy. On the contrary, D3KO tumors have a high ZEB-1/E-cadherin ratio and are highly invasive (Fig. 6).

In conclusion, in this study we addressed the central question of how TH influences different phases of tumorigenesis. Our data for the first time link in vitro mechanistic studies with a model of in vivo carcinogenesis in mice, and, importantly also the results of the analysis of human tumors in which all the hypotheses raised based on in vitro studies are confirmed. By altering the TH signal via cell-specific deiodinase knock-down, we demonstrate that modulation of TH concentration can delay tumor cell growth and invasion depending on tumor stage, and can affect the malignant epithelial tumor phenotype. Therefore, we conclude that D2 is an endogenous "metastasis promoter" and that D2 inhibition can help to reduce human cancer metastasis. These findings not only transform our understanding of how T3 influences tumor progression, but also provides the rationale for the concept that pharmacologically induced TH inactivation could be a strategy with which to attenuate metastatic formations.

## Methods

**Cell cultures and transfections.** SCC-13 cells (Cellosaurus:RRID:CVCL_4029) were derived from skin squamous cell carcinoma (NCIt: C4819), SCC 011 (Cellosarus: RRID:CVCL_5986) were derived from laryngeal squamous cell carcinoma (NCIt: C4044) and SCC 022 (Cellosaurus: RRID:CVCL_5991) were derived from laryngeal squamous cell carcinoma (NCIt: C4044). All the cells were mycoplasma free. SCC-13 and SCC D2KO cells were cultured in keratinocyte-SFM (KSFM 1×) serum-free medium [+] L-Glu (Gibco Life Technologies) with bovine pituitary extract (30 µg/ml) and human recombinant epidermal growth factor (EGF) protein (0.24 ng/ml). SCC D3KO cells were cultured in KSFM with 4% Ca²⁺-chelated charcoal-stripped fetal bovine serum (FBS) and EGF human recombinant (Gibco Life Technologies). All transient transfections were performed using Lipofectamine 2000 (Life Technologies) according to the manufacturer's instructions.

**Luciferase (Luc) expression assays.** The reporter plasmids (T3TRETk-Luc) and CMV-Renilla were co-transfected into SCC-13 cells. Luc activities were measured 48 h after transfection with the Dual Luc Reporter Assay System (Promega), and differences in transfection efficiency were corrected relative to the level of Renilla Luc. Each construct was studied in triplicate in at least three separate transfection experiments.

**Dio2 and Dio3 targeted mutagenesis.** Targeted mutagenesis of Dio3 and Dio2 in SCC was achieved by using the CRISPR/Cas9 system from Santa Cruz Biotechnology. Control SCC cells were stably transfected with the CRISPR/Cas9 control plasmid[10]. Three days after transfection with CRISPR/Cas9 plasmids, the cells were sorted using fluorescence-activated cell sorting (FACS) for green fluorescent protein expression. Single clones were analyzed by PCR to identify alterations in coding regions, and Dio2 exon 1 was sequenced to identify the inserted mutations. All the experiments in D2- and D3KO cells were repeated in three different D3KO and three different D2KO clones to avoid off-target effects.

**Short hairpin RNA-mediated knockdown of ZEB-1.** SCC 13 cells were grown in p60 plates until they reached 60% confluence and then transfected with shRNA (targeting endogenous ZEB-1 or a non-targeting negative control using lipofectamine (Invitrogen)). All the shRNA constructs were obtained from Thermo Fisher

Scientific. Forty-eight hours after transfection, total protein lysate was collected and analyzed by Western blotting or total RNA was extracted and analyzed by real-time PCR. In parallel experiments, 24 h post-transfection, shRNA-transfected cells were used for scratch wound assays.

**Real-time PCR**. Cells and tissues were lysed in Trizol (Life Technologies Ltd.) according to the manufacturer's protocol. 1 μg of total RNA was used to reverse transcribe cDNA using Vilo reverse transcriptase (Life Technologies Ltd.), followed by real-time qPCR using iQ5 Multicolor Real Time Detector System (BioRad) with the fluorescent double-stranded DNA-binding dye SYBR Green (Biorad).

Specific primers for each gene were designed to generate products of comparable sizes (about 200 bp for each amplification). For each reaction, standard curves for reference genes were constructed based on six four-fold serial dilutions of cDNA. All experiments were run in triplicate.

The gene expression levels were normalized to cyclophilin A and calculated as follows: $N \text{*target} = 2(DCt \text{ sample} - DCt \text{ calibrator})$. Primer sequences are indicated in the Supplementary Table 1.

**Protein extraction from skin and western blot analysis**. Dorsal skin was removed from mice and immediately snap-frozen in liquid $N_2$. 800 μl of lysis buffer (0.125 M Tris pH 8.6; 3% SDS, protease inhibitors including PMSF 1 mM and phosphatase inhibitors) were added to all dorsal skin samples, which were then homogenized with Tissue Lyser (Qiagen). Total tissues protein or cell protein was separated by 10% SDS–PAGE followed by Western Blot. The membrane was then blocked with 5% non-fat dry milk in PBS, probed with anti-E-cadherin, anti-N-cadherin, anti-αFlag M2, anti-vimentin, anti-D3, and anti-tubulin antibodies (loading as control) overnight at 4 °C, washed, and incubated with horseradish peroxidase-conjugated anti-mouse immunoglobulin G secondary antibody (1:3000). Band detection was performed using an ECL kit (Millipore, cat. WBKLS0500). The gel images were analyzed using ImageJ software and all Western blot were run in triplicate. Antibodies are indicated in the Supplementary Table 2.

**Wound scratch assay**. SCC CTR, D2KO, and D3KO cells were seeded in p60 plates until they reached 100% confluence. Cells were then treated with mitomycin C from *Streptomyces caespitotus* (0.5 mg/ml). At time T0, a cross-shaped scratch was made on the cell monolayer with the tip of a sterile 2 μl micropipette. The FBS-free culture medium was then replaced with fresh medium to wash out released cells. Cell migration was measured by comparing pictures taken at the beginning and the end of the experiment at the times indicated in each experiment using ×10 magnification with a IX51 Olympus microscope and the Cell*F Olympus Imaging Software. ImageJ software was used to draw the cell-free region limits in each case. The initial cell-free surface was subtracted from the endpoint cell-free surface and plotted in a graph as shown in Fig. 5g.

**Colony formation assay**. To evaluate colony formation, cells were seeded out in appropriate dilutions to form colonies. Five days after plating, cells were washed with PBS and stained with 1% crystal violet in 20% ethanol for 10 min at room temperature. Cells were washed twice with PBS and colonies were counted.

**Invasion assay**. Matrigel chambers (Corning) were used to determine the effect of D3 depletion on invasiveness as per the manufacturer's protocol. In brief, SCC CTR, D2KO, and D3KO cells treated with T3 (30 nM) were harvested, re-suspended in serum-free medium, and then transferred to the hydrated Matrigel chambers (200,000 cells per well). The chambers were then incubated for 5 days in culture medium. The cells on the upper surface were scraped off and washed away, whereas the invaded cells on the lower surface were fixed and stained with 1% crystal violet in 20% ethanol for 10 min at room temperature. Finally, invaded cells and migrated cells were counted under a microscope and the relative number was calculated.

**Matrix metalloproteinase (MMP) assays**. The concentrations of MMP-2, MMP-3, MMP-7, and MMP-13 in the supernatant of SCC-CTR, SCC-CRISPD2, and SCC-CRISPD3 cells were detected via enzyme-linked assays (ELISA) according to the manufacturer's instructions (Cat. nos. ab100603; ab100606; ab100607; ab100608; ab100605). The absorbance at 450 nm was recorded using the VICTOR Multilabel Plate Reader. The general activity of the MMP enzyme was determined using an assay kit purchased from Abcam (Cat. no. ab112146) according to the manufacturer's protocol. In brief, SCC-CTR, SCC-CRISPD2, and SCC-CRISPD3 were seeded into triplicate wells of six-well plates and allowed to attach overnight and then starved with serum-free media for another 18 h. MMP activity was assayed in the media; 25 μl of medium was removed and added to 25 μl of 2 mM APMA working solution and incubated for 15 min at 20 °C after which 50 μl of the green substrate solution was added. An end-point measurement was then performed for the MMP activity using a microplate reader with a filter set of Ex/Em = 490/525 nm.

**Chromatin immunoprecipitation (ChIP) assay**. ChIP Assays were performed in SCC13 cells after treatment with T3 and T4 (30 nM) for 48 h. Briefly, the cells were fixed with 1% formaldehyde for 10 min at 37 °C. The reaction was quenched by the addition of glycine to a final concentration of 0.125 M. After, cells were resuspended in 1 ml of lysis buffer containing protease inhibitors (200 mM phenylmethylsulfony fluoride (PMSF), 1 μg/ml aprotinin) and sonicated to generate chromatin fragments of 200–1000 bp. An aliquot (1/10) of sheared chromatin was treated as Input DNA. Then, the sonicated chromatin was pre-cleared for 2 h with 1 μg of non-immune IgG (Calbiochem) and 30 μl of Protein G Plus/ Protein A Agarose suspension (Calbiochem) saturated with salmon sperm (1 mg/ml). After precleared chromatin was incubated at 4 °C for 16 h with 1 μg of anti-TRα. Protein/DNA cross-links were reversed by incubation in 200 mM NaCl at 65 °C. Finally, DNA extraction was performed with phenol–chloroform and precipitated with ethanol. DNA fragments were used for real-time PCRs.

**Tissue thyroid hormone levels**. Tumors T3 concentration was determined in frozen tissue with an LC–MS/MS method. The iodothyronines were measured with reversed phase chromatography (Waters BEH C18 column: 130 Å, 1.7 μm, 2.1 mm × 50 mm) on an Acquity UPLC-Xevo TQ-S tandem mass spectrometer system (Waters, Milford, MA, USA). Mobile phase A was acetonitrile:$H_2O$:acetic acid 5:95:0.1, mobile phase B was acetonitrile:$H_2O$:acetic acid 95:5:0.1. A gradient was applied with initial percentage B of 10%, which was increased linearly after 2.7 min to 40% in 5.6 min. Total runtime was 12 min. TH and internal standards were quantified using specific MRM transitions. All aspects of system operation and data acquisition were controlled using MassLynx, version 4.1 software with automated data processing using the TargetLynx Application Manager (Waters).

**Animals, histology, and immunostaining**. sD3KO (K14Cre$^{ERT}$-D3$^{fl/fl}$) and sD2KO (K14Cre$^{ERT}$-D2$^{fl/fl}$) mice were obtained by crossing the keratinocyte-specific conditional K14Cre$^{ERT}$ mouse[20] with D3$^{fl/fl}$[9] or D2$^{fl/fl}$ mice[19]. Depletion was induced by treatment with 10 mg of tamoxifen at different time points as indicated in each experiment. The generation of D2-3xFlag is described elsewhere[15]. Skin lesions were harvested at different time points after tamoxifen administration and DMBA-TPA treatment. For immunofluorescence and histology, dorsal skin from sD2KO, sD3KO, D2-3xFlag, and control mice were embedded in paraffin, cut into 7-μm sections, and H&E-stained. Slides were baked at 37 °C, deparaffinized by xylenes, dehydrated with ethanol, rehydrated in PBS, and permeabilized by placing them in 0.2% Triton X-100 in PBS. Antigens were retrieved by incubation in 0.1 M citrate buffer (pH 6.0) or 0.5 M Tris buffer (pH 8.0) at 95 °C for 5 min. Sections were blocked in 1% BSA/0.02% Tween/PBS for 1 h at room temperature. Primary antibodies were incubated overnight at 4 °C in blocking buffer and washed in 0.2% Tween/PBS. Secondary antibodies were incubated at room temperature for 1 h, and washed in 0.2% Tween/PBS. Images were acquired with an IX51 Olympus microscope and the Cell*F Olympus Imaging Software.

**Patients and human tissue samples**. Formalin-fixed, paraffin-embedded tissue blocks of 72 cSCCs, diagnosed and excised from May 1993 to May 2018, were retrieved from the archives of the Pathology Section of the Department of Advanced Biomedical Sciences, 'Federico II' University of Naples. Out of 72 cases, 50 males and 22 females, the age at diagnosis ranged between 36 and 95 years (mean age 76, median 77). Tumor staging (T1–T4) according to the 8th AJCC classification was registered only for cSCCs located on head and neck skin and lip. cSCCs developing on mucosal surfaces and <1 cm in size were not included in this study. The clinical data and pathological features of the tumors are reported in Supplementary Data File 2. The study design and procedures involving tissue samples collection and handling were performed according to the Declaration of Helsinki, in agreement with the current Italian law, and to the Institutional Ethical Committee guidelines.

For immunohistochemistry of human samples, we selected a block of fixed tissue in each case, and one section of the block was stained with hematoxylin/eosin to verify the initial diagnosis; the other sections were used for immunohistochemical investigations.

Immunohistochemical analysis was performed on 4-μm-thick serial sections mounted on poly-L-lysine-coated glass slides. The sections were deparaffinized and subjected to antigen retrieval by microwave oven treatment (3 min × 4 times, in citrate buffer, pH 6); the backdrop (for blocking non-specific background staining) was removed using the universal blocking serum (Dako Diagnostics) for 15 min at room temperature. Endogenous alkaline phosphatase activity was quenched by adding levamisole to slides (substrate buffer); the slides were rinsed with TRIS + Tween20 pH 7.4 buffer, and incubated in a humidified chamber with primary antibodies anti-E-cadherin (mouse monoclonal antibody, diluted 1:300 1 h at room temperature, BD Biosciences), anti-ZEB-1 (rabbit polyclonal antibody, diluted 1:200 1 h at room temperature, NBP1-05987 Novus Biologicals, UK) and anti-D3. Then used a biotinylated secondary antibody and streptavidin conjugated with alkaline phosphatase. The reaction was visualized with chromogen fast red, which showed the presence of the antigen that we sought in red (Dako REAL Detection System, Alkaline Phosphatase/RED, Rabbit/Mouse). After weak nuclear counterstaining with hematoxylin, the sections were mounted with a synthetic medium (Entellan, Merck). Positivity for E-cadherin and ZEB-1 was visualized as red nuclear staining and red membranous staining, respectively. The level of immunostaining was scored semiquantitatively.

**DMBA/TPA carcinogenesis**. The dorsal skin of 2 months old mice was treated with a single dose (100 μl, 1 mg/ml) of the carcinogen 7,12-dimethylbenz[a] anthracene (DMBA) resuspended in propanone, followed by multiple applications of the tumor promoter 12-O-tetradecanoylphorbol-13-acetate, TPA (150 μl, 100 μM) in the two-stage protocol. Experiments in D3Fl/Fl mice were performed using 12 CTR mice and 12 sD3KO mice. Experiments in D2Fl/Fl mice were performed using a total of 30 CTR mice and 30 sD2KO mice. Experiments in D2-Flag mice were performed using eight mice. In all the cohorts there were approximately as many female as male mice. As shown in Fig. S13, the number of metastatic organs was evaluated based on the overall expression of K8 and K14.

**Isolation of TICs and transplantation in immunodeficient mice**. Papillomas and carcinomas arising from DMBA/TPA D2-3xFlag mice were digested in collagenase I (Sigma) for 2 h at 37 °C on a rocking plate. Collagenase I activity was blocked by the addition of DMEM with 10% FBS. After tumor digestion, cells were filtered through a 70 μm cell strainer. Immunostaining was performed using APC-anti-mouse CD34 (code 119310; Biolegend), PE-rat anti-human α6-integrin (code 555736; BD Pharmingen), FITC anti-CD3/GR-1/CD11B/CD45-B220/TER 119 (code 78022; Biolegend), vioBlu-anti mouse CD326 (code 130-102-421; Milteny Biotech) by incubation for 1 h. FACS analysis was performed using FACS Canto2 software (FACS Canto2, Becton Dickinson). Sorted cells were collected for in vivo transplantation experiments. In detail, cells were resuspended in 8% FBS with matrigel (100 μL [8 mg/ml] Sigma) and injected subcutaneously into athymic null mice. Triplicate injections per mouse were performed. Tumor volume was calculated with the formula $V \frac{1}{4} p [d2 D]/6$, where $d$ is the minor tumor axis and $D$ is the major tumor axis. All animal experiments and mouse husbandry were carried out in the animal facility of CEINGE-Biotecnologie Avanzate, Naples, Italy, in accordance with institutional guidelines.

**Animal study approval**. All animal studies were conducted in accordance with the guidelines of the Ministero della Salute and were approved by the Institutional Animal Care and Use Committee (IACUC, nos. 167/2015-PR and 354/2019-PR).

**Statistics**. The data are expressed as means ± standard deviation (SD) of three independent experiments. Statistical differences and significance between samples were determined using the Student's two-tailed $t$ test. Relative mRNA levels (in which the control sample was arbitrarily set as 1) are reported as results of real-time PCR, in which the expression of cyclophilin A served as housekeeping gene. A $P$ value < 0.05 was considered statistically significant. Asterisks indicate significance at *$P$ < 0.05, **$P$ < 0.01, and ***$P$ < 0.001 throughout.

**Reporting summary**. Further information on research design is available in the Nature Research Reporting Summary linked to this article.

## Data availability
The data that support the findings of this study are available from the corresponding author upon reasonable request.

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

## Acknowledgements
This work was supported by the grant from AIRC to M.D. (IG 13065) and by the grant from AIRC to C.M. (IG 17079), by the ERCStG2014 grant from the European Research Council under the European Union's Horizon2020 Program (STARS—639548) and by the grant FARE from the Ministero dell'Istruzione, dell'Università e della Ricerca-MIUR-R16KPLYF38) to M.D.; by the grant from the European Research Council under the European Union's Horizon2020 Programme—EU FP7 contract Thyrage (grant number

666869) to D.S. and M.D. We thank Jean Ann Gilder (Scientific Communication srl., Naples, Italy) for editing the text.

## Author contributions

C. Miro, E.D.C., D.D.G., A.G.C., S. Sagliocchi, A.N., and C.L. performed the in vitro and in vivo experiments and prepared the figures; R.A. generated the plasmids and mouse models; G.M. and M.A.D.S. performed the histochemistry and immunofluorescence analysis; D.A. conduced the bioinformatics analysis; F.V. and L.D.V performed the FACS analysis studies; S.V. and G.I. collected the human tumor samples and performed histochemistry; A.B. performed the intra-tumoral TH measurements; A.B., S. Staibano, C.B., C. Missero, and D.S. provided observations and scientific interpretations; D.S. contributed to experiment supervision and interpretation; M.D. designed the overall study, supervised the experiments, analyzed the results, and wrote the paper; all authors discussed the results and provided input on the manuscript.

## Competing interests

The authors declare no competing interests.
