## [Peer Review File · Nature Communications]

Reviewers' comments:

Reviewer #1 (Expertise: Thyroid hormone, cancer, Remarks to the Author):

Miro and co-workers implicate T3 (and the action of DIO¹ or D2) in hormonal support for SCC cell proliferation via actions of T3 on EMT through ZEB-1 and vimentin. This is an interesting and thorough study from a laboratory which has provided an enormous amount of information to us about the functions of deiodinases.

The principal concern here is the high total T3 concentration (30 nM) used in the SCC cell studies. It is likely that 30 nM total T3 concentration generated a supraphysiological level of free hormone in the authors' serum-supplemented cell culture medium. Supraphysiological levels of T3 do cause tumor cell proliferation (see, for example, HY Lin et al., *Am J Physiol Cell Physiol* 296:C980-C991, 2009), but clinically--for example, in the euthyroid hypothyroxinemia state (A Hercbergs, *Oncologist* 20:72-76, 2015)--normal circulating levels of T3 are associated with stabilized or reduced tumor size.

Further, we would expect the nonthyroidal illness syndrome (NTIS) which complicates the courses of serious illness, including cancer, and which is associated with low endogenous T3 levels to lead to improved cancer outcomes. So far, this has not been observed. NTIS is instead associated with worsened prognosis in cancer (SE Cengiz, *Intern Med* 47:211-216, 2008; ZA Yasar, *Horm Cancer* 5:240-246, 2014; R Gao, *Int J Cancer* 14:466-477, 2018), as well as in serious noncancer disease states. The authors mention the therapeutic option of manipulating deiodinase activity in cancer patients to lower blood T3 levels, but this happens spontaneously in the clinical arena.

Reviewer #2 (Expertise: CHIP-seq, Remarks to the Author):

The manuscript by C. Miro et al., focuses on the role of the thyroid hormone (TH) on the evolution of squamous cell carcinoma (SCC). The major claim of the study is that TH attenuates the early evolution of SCC (by inhibiting SCC proliferative capacity), but induces malignant evolution by activating an EMT cascade leading to metastatic dissemination.

The data presented are convincing, they combine in vivo and in vitro models and the authors provide a novel mechanistic link between TH-receptor and direct activation of the master EMT regulator ZEB1. Thereby the paper provides a novel view on the stepwise development of SCC and is of interest to a wide array of scientists interested in cancer biology, EMT and/or TH biology.

Although this study is of very good quality, to my opinion, several points need to be addressed before the article may be formally accepted for publication:

major comments

(i) The authors use two different shRNA directed against ZEB1 to show its requirement for metastasis and SCC cells invasiveness and dissemination (Fig. S12). They show that both shRNA are efficient at knocking down ZEB1. However, in Fig. 6E they use only one shRNA to carry-out the functional assays. It is unclear why, and the authors should show that both sh are producing the same biological effect in order to rule out any potential off target effect.

(ii) along the same line, the authors show that another TH receptor target, Vimentin, is not mediating the invasiveness of SCC (as opposed to ZEB1). It is not clear why, to tackle the same question, the authors use a different readout when dealing with Vimentin? i.e. E-cadherin mRNA quantification only whereas for ZEB1, the authors show a complete panel of experiments including E-cadh/N-cadh western blot and protein quantification, MMP9 mRNA quantification (Fig. 6F, G). More consistency is needed to be more convincing here.

(iii) I honestly do not see the added value of the ChIP-Seq of TH receptor shown in Fig. S10. First, it is not properly depicted, as only the Zeb1 and Vimentin loci contain the actual ChIP-Seq signals (i.e. peaks). For the other loci, only black boxes are shown, probably representing the results of the peak calling algorithm (by the way no information is provided concerning the peak calling or the control dataset used). Also, it is impossible to judge the quality of this ChIP-Seq dataset as absolutely no control track is shown. What has been used? control Ig? input? This would be very useful for instance to judge if the peak present at the Zeb1 locus is really a peak or just some occasional background signal (indeed the shape of the peak is rather strange and looks more like a stack of sequences than a real signal...). Furthermore, this ChIP-Seq is not publicly available: I can understand that the authors may want to use this dataset in another study and want to keep it private, but why showing it then? In addition it comes from an unrelated cell line (C2C12 cells), so it is only poorly indicative of what is actually happening in SCC cells.

So, my suggestion is : 1-to remove this information as it does not add anything useful in the present study. Or if the authors insist to keep it they should depict it correctly and add the relevant controls and information; 2- to test by ChIP-qPCR the proximal promoters of the other genes tested (i.e. TGF- β 2, Col1a2, etc.). If negative, these data may be kept as supporting information. Once again, the use of the ChIP-Seq guided the authors towards the putative direct target genes of TH receptor but the author could have simply indicated that they checked by ChIP-qPCR a select set of targets and found that both ZEB1 and Vimentin scored positive.

Minor comments:

(i) the terminology used to label ZEB1 shRNAs is misleading. The authors should use shZEB1-1 and shZEB1-2 (instead of ZEB-1 and ZEB-2) to avoid mixing-up "ZEB-2" with Zeb2 which is another EMT-related factor.

(ii) the protein levels and ratios of E-cadherin versus N-cadherin shown in Fig.5A and Fig.6F are not extremely convincing (whereas the quantification is fine). Is it possible to show a different exposure or to show a better western blot analysis?

(iii) on page 22, line 697, the authors refer to Figure S8, but it should be S10.

(iv) in the introduction, page 3 line 64, the authors should replace '(ref)' by the actual relevant reference.

As summary, this is a convincing study with a nice complementary array of experiments that support the claims.

Reviewer #3 (Expertise: SCC, Remarks to the Author):

Comments: The authors explored the effects of TH signaling and its regulators in the late stages of the neoplastic process in squamous cell carcinoma (SCC) by using the classical two-step carcinogenesis model in mouse skin and in vitro cell model. They found that the D2-mediated T3 production promotes the transcription of ZEB-1 and vimentin, which enables activation of the epithelial-mesenchymal transition (EMT) cascade, thus resulting in enhanced invasiveness of SCC cells. In conclusion, D2 is an endogenous "metastasis promoter" and that D2-inhibition can help to reduce human cancer metastasis.

Overall, these are interesting findings and the conclusions are generally supported by the data presented. The authors have performed comprehensive in vivo experiments to provide evidence for their findings.

However, as listed below there are a number of weaknesses and concerns which need to be addressed to further strengthen their findings. In general, the in vitro and molecular investigations are not in-depth and lack mechanistic insights. The manuscript was not well written and needs systematic polishing.

1. As the molecular mechanisms underlying the regulation of TH on EMT is a key point of this manuscript, it is hard to justify that the authors chose not to show their Chip-seq data of thyroid hormone receptor binding sites, which was mentioned as “unpublished data” in line 248. I think it is absolutely necessary from a research point of view to upload these Chip-seq data.
2. 8 genes (ZEB-1, TGF- β 2, N-cadherin, vimentin, Col1a2, 250 Serpine-1, SNAI2 and MAP-1B) contained multiple T3-binding sites based on Chip-seq but only ZEB-1 and vimentin could be validated by qPCR? This low validation rate is surprising, which calls to question the Chip-seq data quality and analytic methods. For example, how many peaks in total are identified and where are they distributed? Does T3-binding show cell type specificity, i.e., does it bind differently in different cell types?
3. How does T3 regulate ZEB-1 and vimentin transcription? By enhancer or promoter activation?
4. The authors suppressed D2 or D3 expression in SCC cells using CRISPR/Cas9 technology to explore the role of the intracellular control of TH action in EMT. This conclusion needs to be further confirmed by overexpressing D2 or D3 in the SCC cell lines.
5. They analyzed the expression patterns of a panel of eight metalloproteases (MMP-2, 3, 7, 8, 9, 10 and 13) in D3KO and D2KO cells by determining the mRNA levels of these metalloproteases using Real time PCR. However, the activities of these proteases are more important than the expression levels of them. Thus it would be more convincing if the proteases activity can be measured
6. In the survival correlation section, it is not clear if the prognostic value of D2 is statistically significant. In addition, since the authors was correlating the mRNA levels, they should also check other publicly datasets, particularly TCGA which has hundreds of SCC samples with survival data. Their own 50 patients is a rather small cohort.
7. The manuscript is not well written. For example, the 1st sentence of introduction, what is “(ref)”? also ,the 1st sentence mentions hormonal regulation of carcinogenesis, but the rest of paragraph 1 has nothing to do with hormone. Another example, the 1st sentence of paragraph 2 “Thyroid hormones (TH) T4 and T3 regulate the metabolism and growth of all cell types, and thereby have a strong impact on cancer”. It is a very strong statement, but is it true that “all cell types” have been tested? It would be a rather careless and reckless statement if the authors have not verified.

Reviewer #4 (Expertise: EMT, metastasis, Remarks to the Author):

In this manuscript, the authors investigated how Th signals influence different phases of tumorigenesis. They found D2-inhibition can help to reduce human cancer metastasis, while D3-deficiency promotes tumor initiation and decreases metastasis. The expression of D2 was increased in the late stage of tumorigenesis, while D3 was highly expressed until the formation of papillomas. In general, this manuscript delivered an interesting story. The authors used a novel in vivo multi-step carcinogenesis mice model together with cell-specific deiodinase knock-down to observe the dynamic expression of D2 and D3, and their functions on tumor cell growth and invasion.

Some issues need to be addressed.

1. It seems the determinant factor on tumor cell growth or metastasis is the level of T3. However, the authors didn't show the dynamic T3 concentrations during tumor formation or progression.
2. The authors concluded that D2 is an endogenous "metastasis promoter" and analyzed the clinical significance of D2. However, deficiency of D3 had the same effect with D2, as shown in figure 2, knockdown of D3 accelerated tumor metastasis even in the tumor promotion stage. Therefore, the authors should also detect the expression of D3 in tumor metastases and analyze its clinical significance.
3. The authors should increase clinical sample size. For 7C and D, n=10 was not enough. Also, the authors should analyze clinical data to show the relationship between the expression of D2 or D3 with tumor grade.
4. For all the EMT-related WB experiments, the changes of E-cadherin or N-cadherin were subtle. However, immunofluorescence and cell morphology showed a very clear E or M phenotype. The authors should optimize their WB experiments or detect other EMT markers.
5. In fig 5H a, the representative images were not consistent with the statistical column.

ANSWERS TO THE REVIEWERS

We would like to thank the reviewers for their valuable and insightful comments. All comments were addressed point by point and by the following additional figures.

Reviewer #1 (Expertise: Thyroid hormone, cancer, Remarks to the Author):

Miro and co-workers implicate T3 (and the action of DIO¹ or D2) in hormonal support for SCC cell proliferation via actions of T3 on EMT through ZEB-1 and vimentin. This is an interesting and thorough study from a laboratory which has provided an enormous amount of information to us about the functions of deiodinases.

The principal concern here is the high total T3 concentration (30 nM) used in the SCC cell studies. It is likely that 30 nM total T3 concentration generated a supraphysiological level of free hormone in the authors' serum-supplemented cell culture medium. Supraphysiological levels of T3 do cause tumor cell proliferation (see, for example, HY Lin et al., Am J Physiol Cell Physiol 296:C980-C991, 2009), but clinically--for example, in the euthyroid hypothyroxinemia state (A Hercbergs, Oncologist 20:72-76, 2015)--normal circulating levels of T3 are associated with stabilized or reduced tumor size.

A- We thank the reviewer for raising this issue. We performed our studies in culture media containing 10% FBS, and endogenous hormone binding proteins present in the serum. Consequently, the effective free T3 concentration in the culture media -when we used 30 nM T3- was about 3nM, a dose that is ten times the physiological T3 concentrations in human plasma (1.2-3.4 nmol/L) but still compatible with the elevated T3 observed in hyperthyroid patients.

The function of T3 and TRs in neoplastic cell proliferation involves complex mechanisms that seem to be cell specific, via genomic and non-genomic pathways, and represses or stimulates cell growth, angiogenesis and invasiveness in different tumors.

The general concept in this scenario is that the overall effects of T3 on the proliferation/differentiation balance are heterogeneous and highly time and context dependent. In our case, we observed that T3 differently affects tumoral growth depending on the specific tumoral stage.

Furthermore, the surge of T3 that we observed in advanced SCC did not affect systemic levels of thyroid hormone, but it was a local, intra-tumoral adaptation, that regulates the growth and evolution of tumoral cells.

Further, we would expect the nonthyroidal illness syndrome (NTIS) which complicates the courses of serious illness, including cancer, and which is associated with low endogenous T3 levels to lead to improved cancer outcomes. So far, this has not been observed. NTIS is instead associated with worsened prognosis in cancer (SE Cengiz, Intern Med 47:211-216, 2008; ZA Yasar, Horm Cancer 5:240-246, 2014; R Gao, Int J Cancer 14:466-477, 2018), as well as in serious noncancer disease states. The authors mention the therapeutic option of manipulating deiodinase activity in cancer patients to lower blood T3 levels, but this happens spontaneously in the clinical arena.

*A- As the reviewer discussed, the overall effects of systemic T3 levels on cancer formation and progression are still a matter of debate and highly variable in different studies and in different patients. The therapeutic implications of our study include the possibility to hamper cancer progression by deiodinase manipulation **customized to the specific tumour stage**, which can lead to local T3 modulation of intracellular T3 concentrations without affecting the T3 levels in the plasma. Of course strategies to target specifically deiodinases within the tumor compartment are not available yet, but our data suggest that targeting intra-tumoral D2 can represent a valid option with which to interfere with SCC tumor progression.*

Reviewer #2 (Expertise: ChIP-seq, Remarks to the Author):

The manuscript by C. Miro et al., focuses on the role of the thyroid hormone (TH) on the evolution of squamous cell carcinoma (SCC). The major claim of the study is that TH attenuates the early evolution of SCC (by inhibiting SCC proliferative capacity), but induces malignant evolution by activating an EMT cascade leading to metastatic dissemination.

The data presented are convincing, they combine in vivo and in vitro models and the authors provide a novel mechanistic link between TH-receptor and direct activation of the master EMT regulator ZEB1. Thereby the paper provides a novel view on the stepwise development of SCC and is of interest to a wide array of scientists interested in cancer biology, EMT and/or TH biology.

Although this study is of very good quality, to my opinion, several points need to be addressed before the article may be formally accepted for publication:

major comments

The authors use two different shRNA directed against ZEB1 to show its requirement for metastasis and SCC cells invasiveness and dissemination (Fig. S12). They show that both shRNA are efficient at knocking down ZEB1. However, in Fig. 6E they use only one shRNA to carry-out the functional assays. It is unclear why, and the authors should show that both sh are producing the same biological effect in order to rule out any potential off target effect.

A- To address this point, we performed the experiments shown in Fig. 6 with the second shZEB1 and the effects on cell migration and epithelial-mesenchymal transition were confirmed. The new data are shown in Fig. S14 and described in the Results (page 8). Similarly, for Vimentin knock-down, we report the data obtained with the second shVimentin (Additional Fig. 1).

along the same line, the authors show that another TH receptor target, Vimentin, is not mediating the invasiveness of SCC (as opposed to ZEB1). It is not clear why, to tackle the same question, the authors use a different readout when dealing with Vimentin? i.e. E-cadherin mRNA quantification only whereas for ZEB1, the authors show a complete panel of experiments including E-cadh/N-cadh western blot and protein quantification, MMP9 mRNA quantification (Fig. 6F, G). More consistency is needed to be more convincing here.

A- We obtained more robust data that confirmed that Vimentin is a TH-target gene, although vimentin silencing is not sufficient to block the effects of TH on EMT (Additional Fig. 1).

I honestly do not see the added value of the ChIP-Seq of TH receptor shown in Fig. S10. First, it is not properly depicted, as only the Zeb1 and Vimentin loci contain the actual ChIP-Seq signals (i.e. peaks). For the other loci, only black boxes are shown, probably representing the results of the peak calling algorithm (by the way no information is provided concerning the peak calling or the control dataset used). Also, it is impossible to judge the quality of this ChIP-Seq dataset as absolutely no control track is shown. What has been used? control Ig? input? This would be very useful for instance to judge if the peak present at the Zeb1 locus is really a peak or just some occasional background signal (indeed the shape of the peak is rather strange and looks more like a stack of sequences than a real signal...). Furthermore, this ChIP-Seq is not publicly available: i can understand that the authors may want to use this dataset in another study and want to keep it private, but why showing it then? In addition it comes from an unrelated cell line (C2C12 cells), so it is only poorly indicative of what is actually happening in SCC cells.

So, my suggestion is : 1-to remove this information as it does not add anything useful in the present study. Or if the authors insist to keep it they should depict it correctly and add the relevant controls and information; 2- to test by ChIP-qPCR the proximal promoters of the other genes tested (i.e. TGF-b2, Col1a2, etc.). If negative, these data may be kept as supporting information. Once again, the use of the ChIP-Seq guided the authors towards the putative direct target genes of TH receptor but the author could have simply indicated that they checked by ChIP-qPCR a select set of targets and found that both ZEB1 and Vimentin scored positive.

A- We take the reviewer's point and removed the ChIP-Seq data from the manuscript. This decision was due to different considerations: first, the old ChIP-seq was performed in a cellular model different from the cellular model used in our manuscript as both the reviewer 2 and 3 pointed out. Second, regarding the possibility to up-load the old ChIP-seq (performed in the 2009), this is unfortunately not possible since we have only one wig file. Third, we performed a new ChIP-seq on SCC cells using two different antibodies against TR β and TR α (the two thyroid receptors expressed in the skin and in SCC). However, the TR α antibody used to perform the old ChIP-seq (abcam, cod. ab2743) is not available anymore in commerce and the new one that we used (abcam, cod. ab5622) generated high background in the sequencing process following ChIP-Seq. Therefore, due to the absence of alternative ChIP-grade antibodies available at the moment, we followed the first reviewer suggestion and removed the ChIP-seq data.

In the revised version of the manuscript, in order to gain insights the molecular determinants regulating EMT under the control of TH, we focused our studies on ZEB-1 as the master regulator of EMT and up-stream regulator of EMT related genes. We analyzed the 5' flanking region (2000 bp up- and 2000 bp-down stream the TSS) of the human ZEB-1 gene. We searched for thyroid hormone binding elements (TRE binding sites) within the promoter and found different TREs in the 5'-FR of ZEB-1. Finally, by using the last microliters of the ChIP-grade antibody (abcam, cod. ab2743) we performed additional ChIP analysis, which confirmed the direct binding of T3-TR on

ZEB-1 promoter by three different replicate experiments and validated the specificity of the data using an unrelated DNA region as negative control (Fig. 6). All together, these analyses confirmed ZEB1 a positive, direct target of T3.

Minor comments:

the terminology used to label ZEB1 shRNAs is misleading. The authors should use shZEB1-1 and shZEB1-2 (instead of ZEB-1 and ZEB-2) to avoid mixing-up "ZEB-2" with Zeb2 which is another EMT-related factor.

A- We thank the reviewer. We have corrected the terminology throughout the manuscript.

the protein levels and ratios of E-cadherin versus N-cadherin shown in Fig.5A and Fig.6F are not extremely convincing (whereas the quantification is fine). Is it possible to show a different exposure or to show a better western blot analysis?

A- We have repeated the western blots in Fig. 5A and 6F and now the results are more clear and -we hope-convincing.

on page 22, line 697, the authors refer to Fig. S8, but it should be S10.

A- We thank the reviewer and Fig. S8 now reads Fig. S10.

in the introduction, page 3 line 64, the authors should replace '(ref)' by the actual relevant reference.

A- This sentence has been deleted as suggested by reviewer 3.

As summary, this is a convincing study with a nice complementary array of experiments that support the claims.

Reviewer #3 (Expertise: SCC, Remarks to the Author):

Comments: The authors explored the effects of TH signaling and its regulators in the late stages of the neoplastic process in squamous cell carcinoma (SCC) by using the classical two-step carcinogenesis model in mouse skin and in vitro cell model. They found that the D2-mediated T3 production promotes the transcription of ZEB-1 and vimentin, which enables activation of the epithelial-mesenchymal transition (EMT) cascade, thus resulting in enhanced invasiveness of SCC cells. In conclusion, D2 is an endogenous "metastasis promoter" and that D2-inhibition can help to reduce human cancer metastasis.

Overall, these are interesting findings and the conclusions are generally supported by the data presented. The authors have performed comprehensive in vivo experiments to provide evidence for their findings.

However, as listed below there are a number of weaknesses and concerns which need to be

addressed to further strengthen their findings. In general, the in vitro and molecular investigations are not in-depth and lack mechanistic insights. The manuscript was not well written and needs systematic polishing.

As the molecular mechanisms underlying the regulation of TH on EMT is a key point of this manuscript, it is hard to justify that the authors chose not to show their Chip-seq data of thyroid hormone receptor binding sites, which was mentioned as “unpublished data” in line 248. I think it is absolutely necessary from a research point of view to upload these Chip-seq data.

A- We take the reviewer’s point and removed the ChIP-Seq data from the manuscript. This decision was due to different considerations: first, the old ChIP-seq was performed in a cellular model different from the cellular model used in our manuscript as both the reviewer 2 and 3 pointed out. Second, regarding the possibility to upload the old ChIP-seq (performed in the 2009), this is unfortunately not possible since we have only one wig file. Third, we performed a new ChIP-seq on SCC cells using two different antibodies against TR β and TR α (the two thyroid receptors expressed in the skin and in SCC). However, the TR α antibody used to perform the old ChIP-seq (abcam, cod. ab2743) is not available anymore in commerce and the new one that we used (abcam, cod. ab5622) generated high background in the sequencing process following ChIP-Seq. Therefore, due to the absence of alternative ChIP-grade antibodies available at the moment, we followed the reviewer’s 2 suggestion i.e. “to remove this information as it does not add anything useful in the present study”.

In the revised version of the manuscript, in order to gain insights the molecular determinants regulating EMT under the control of TH, we focused our studies on ZEB-1 as the master regulator of EMT and up-stream regulator of EMT related genes. We analyzed the 5’ flanking region (2000 bp up- and 2000 bp-down stream the TSS) of the human ZEB-1 gene. We searched for thyroid hormone binding elements (TRE binding sites) within the promoter and found different TREs in the 5’-FR of ZEB-1. Finally, by using the last microliters of the ChIP-grade antibody (abcam, cod. ab2743) we performed additional ChIP analysis, which confirmed the direct binding of T3-TR on ZEB-1 promoter by three different replicate experiments and validated the specificity of the data using an unrelated DNA region as negative control (Fig. 6).

All together, these analyses confirmed ZEB1 a positive, direct target of T3.

8 genes (ZEB-1, TGF- β 2, N-cadherin, vimentin, Col1a2, 250 Serpine-1, SNAI2 and MAP-1B) contained multiple T3-binding sites based on Chip-seq but only ZEB-1 and vimentin could be validated by qPCR? This low validation rate is surprising, which calls to question the Chip-seq data quality and analytic methods. For example, how many peaks in total are identified and where are they distributed? Does T3-binding show cell type specificity, i.e., does it bind differently in different cell types?

A- Since we could not repeat the ChIP-seq analysis, we validated the expression of the EMT profiler in Fig. S11B by regular real time PCR analysis and decided to focus our attention on the ZEB-1 regulation.

How does T3 regulate ZEB-1 and vimentin transcription? By enhancer or promoter activation?

A- T3 regulates ZEB-1 transcription by promoter activation. Infact, in silico analysis demonstrated the presence of different TRE binding sites in the 5'-FR of human ZEB-1 gene. In the revised manuscript, we added ChIP analyses showing that TR α specifically binding to this region, thus confirming that T3 regulates ZEB-1 transcription (Fig. 6 and S12).

The authors suppressed D2 or D3 expression in SCC cells using CRISPR/Cas9 technology to explore the role of the intracellular control of TH action in EMT. This conclusion needs to be further confirmed by overexpressing D2 or D3 in the SCC cell lines.

A- We thank the reviewer for raising this point. We transiently overexpressed D2 and D3 in SCC cells and measured mRNA levels of E-cadherin, N-cadherin, Vimentin and Twist were measured. The results of these experiments confirmed that EMT markers were up-regulated in D2-overexpressing cells while D3 overexpression reduced the levels of EMT genes (Fig. S7).

They analyzed the expression patterns of a panel of eight metalloproteases (MMP-2, 3, 7, 8, 9, 10 and 13) in D3KO and D2KO cells by determining the mRNA levels of these metalloproteases using Real time PCR. However, the activities of these proteases are more important than the expression levels of them. Thus it would be more convincing if the proteases activity can be measured

A- To address the reviewer question, we measured both the production and secretion levels of the specific metalloproteases in the culture medium, and the activity of the metalloproteases in the cell lysates. As shown in revised Fig. S10, T3 treatment enhances not only the transcription of the metalloproteases, but also their secretion and enzymatic activity.

In the survival correlation section, it is not clear if the prognostic value of D2 is statistically significant. In addition, since the authors was correlating the mRNA levels, they should also check other publicly datasets, particularly TCGA which has hundreds of SCC samples with survival data. Their own 50 patients is a rather small cohort.

A- We apologize for not including the p value for the association of D2 with the survival rate and the percentage of recurrence, which is now correctly indicated in the Fig. 7. As suggested, we extended our analysis to the TCGA data. Although the TGCA does not contain data on human cutaneous SCC samples, we assessed the correlation between D2 expression and survival rate of head and neck SCC, lung SCC, cervical SCC and esophageal carcinomas.

In the cervical SCC, D2 expression is not significantly up-regulated in primary tumor when compared with normal tissue and there is not a significant correlation between D2 and percent survival;

In head and neck SCC, D2 is down regulated in the primary tumor and there is not a significant correlation between D2 and percent survival;

In the esophageal carcinomas, D2 expression is significantly up-regulated in primary

*tumor when compared with normal tissue and there is also a negative, significant correlation between D2 and percent survival;
In the lung SCC, D2 expression is not significantly up-regulated in primary tumor when compared with normal tissue and there is not a significant correlation between D2 and percent survival;*

We attribute this highly heterogeneous data to the difference among the subtypes of SCCs in terms of oncogene mutations, pathways activation and metabolic changes. The new data are indicated as additional data 2.

Notably, as the reviewer asked, we extended the human samples size to the analysis of 72 patients and the mRNA extracted from the tumors were extended from 10 to 20. The new larger analysis confirmed the previous results, which are now reported in the new version of Fig. 7 and Table 2.

Finally, we to reinforce the clinical relevance of our finding, we have also:

- i. Measured D3 mRNA expression in tumor metastases. As shown in the new Fig. 7B, D3 expression is lower in lymph nodes and metastasis when compared to the normal tissues while D2 and K8 expressions are higher in lymph nodes and metastasis;*
- ii. Measured D3 protein levels in the human SCC panel by immunohistochemical analysis (Fig. 7D). Importantly, we found that D3 is highly expressed in the early stages of tumorigenesis in human SCC (G1) as already shown in mouse SCC (Fig. 1B).*
- iii. Measured D3 mRNA levels in the human SCC panel (Fig. 7D). D3 mRNA expression parallels the protein expression.*
- iv. Assessed the clinical significance of D3 expression in the tumors and its correlation with the Survival and Recurrence rate. As shown in Fig. 7E, D3 expression is positively associated with reduced Survival of patients, thus confirming that alteration of intratumoral T3 worsen the tumor prognosis. At the opposite, the percentage of recurrence is not significantly associated with altered D3 expression.*

The manuscript is not well written. For example, the 1st sentence of introduction, what is "(ref)"? also ,the 1st sentence mentions hormonal regulation of carcinogenesis, but the rest of paragraph 1 has nothing to do with hormone.

A- The paper has been language edited by an author's editor whose first language is English and who is a long-standing member of the European Association of Science Editors. Moreover, we agree with the reviewer and we removed the reported statement that is not in line with the following text.

Another example, the 1st sentence of paragraph 2 "Thyroid hormones (TH) T4 and T3 regulate the metabolism and growth of all cell types, and thereby have a strong impact on cancer". It is a very strong statement, but is it true that "all cell types" have been tested? It would be a rather careless and reckless statement if the authors have not verified.

A- The statement "Thyroid hormones (TH) T4 and T3 regulate the metabolism and growth of all cell types, and thereby have a strong impact on cancer" refers to the

concept that thyroid hormone receptors are expressed ubiquitously and consequently, all the cells of our body are bona fide potential targets of thyroid hormone. However, we have now toned down the sentence as follow: "Thyroid hormones (TH) T4 and T3 are pleiotropic agents that regulate the metabolism and growth of many cells and tissues, and thereby have a strong impact on cancer"

Reviewer #4 (Expertise: EMT, metastasis, Remarks to the Author):

In this manuscript, the authors investigated how Th signals influence different phases of tumorigenesis. They found D2-inhibition can help to reduce human cancer metastasis, while D3-deficiency promotes tumor initiation and decreases metastasis. The expression of D2 was increased in the late stage of tumorigenesis, while D3 was highly expressed until the formation of papillomas. In general, this manuscript delivered an interesting story. The authors used a novel in vivo multi-step carcinogenesis mice model together with cell-specific deiodinase knock-down to observe the dynamic expression of D2 and D3, and their functions on tumor cell growth and invasion.

Some issues need to be addressed.

It seems the determinant factor on tumor cell growth or metastasis is the level of T3. However, the authors didn't show the dynamic T3 concentrations during tumor formation or progression.

A- We thank the reviewer for raising this point that helped improve our study. We have measured the intracellular T3 concentration in the SCC tumors at the stages of Papilloma and advanced SCC. Notably, the T3 levels in the papillomas are significantly lower than that present in the SCC. These data are consistent with the high D3 and low D2 levels in papilloma and the opposite balance observed in SCC.

We are grateful to the reviewer's suggestion, which enabled us to show for the first time that the intra-tumoral levels of the thyroid hormone change in the different stages of tumor progression. The important new finding is now shown in Fig. 1D.

The authors concluded that D2 is an endogenous "metastasis promoter" and analyzed the clinical significance of D2. However, deficiency of D3 had the same effect with D2, as shown in Fig. 2, knockdown of D3 accelerated tumor metastasis even in the tumor promotion stage. Therefore, the authors should also detect the expression of D3 in tumor metastases and analyze its clinical significance.

A- As suggested by the reviewer, we have:

i. Measured D3 mRNA expression in the tumor metastasis. As shown in the new Fig. 7B, D3 expression was lower in metastatic lymph nodes and lung metastasis when compared to the normal corresponding tissues; conversely, D2 and K8 were highly expressed in lymph nodes and lung metastasis;

ii. Measured D3 protein levels in the human SCC panel by immunohistochemical analysis (Fig. 7C). As observed in the mouse panel shown in Fig. 1B, D3 was highly expressed in the early stages of tumorigenesis in human SCC as already shown in mouse SCC (Fig. 1B).

- iii. Measured D3 mRNA levels in the human SCC panel (Fig. 7D). D3 mRNA expression parallels D3 protein expression.*
- iv. Assessed the clinical significance of D3 expression in the tumors and its correlation with the Survival and Recurrence rate. As shown in Fig. 7F, D3 expression is positively associated with reduced Survival of patients, thus confirming that reduced T3 promotes initial tumor growth. The percentage of tumor recurrence is not significantly associated with altered D3 expression.*

The authors should increase clinical sample size. For 7C and D, n=10 was not enough. Also, the authors should analyze clinical data to show the relationship between the expression of D2 or D3 with tumor grade.

A- We have expanded our analysis of human SCC and analyzed 10 new cases (total of 20). The new analysis confirmed the elevated expression of D2 in SCC as shown in Fig. 7E. Moreover, as indicated before, we have measured the expression of D3 mRNA and protein levels in the human tumors at different stages and the new data reinforce the inverse correlation of D2 and D3 with the tumor grading.

For all the EMT-related WB experiments, the changes of E-cadherin or N-cadherin were subtle. However, immunofluorescence and cell morphology showed a very clear E or M phenotype. The authors should optimize their WB experiments or detect other EMT markers.

A- We have repeated the western blots analysis of E-cadherin and N-cadherin in Fig. 5A, 6F and confirmed our results.

In fig 5H a, the representative images were not consistent with the statistical column.

A- We agree with the reviewer and apologize for having miscalculated the percentage of invading cells in Fig. 5H. We have recalculated the value that is actually 24.3%. We have corrected the Figure accordingly.

ADDITIONAL DATA

Additional Figure 1.

(A) Wound scratch assay of SCC cells after exposure to 30 nM T3 or T3 + Vimentin shRNA. Scale bars represent 200 μm . (C) E-cadherin and N-cadherin expression was evaluated by western blot analysis in the same cells as in A. (D) mRNA levels of MMP9 in SCC cells treated or not with 30 nM T3 and transfected with shVimentin or shCTR.

Additional Figure 2. Expression of D2 based on the TCGA database was correlated to percent of survival.

REVIEWERS' COMMENTS:

Reviewer #1 (Remarks to the Author):

The authors have provided additional relevant information about the supraphysiologic levels of T3 that are required to affect EMT via SEB-1 in SCC cells. I remain concerned about the 10-fold higher than physiologic concentrations of T3 used in the studies described. But inclusion by the authors in the revision of the information on this issue provided in their comments to the Reviewers provides to readers a satisfactory perspective on this issue.

Reviewer #2 (Remarks to the Author):

The authors have satisfactorily addressed my points. The manuscript is now improved, and I have no further comments.

Reviewer #3 (Remarks to the Author):

I appreciate that the authors have performed extensive revision experiments and have addressed most of my original concerns. However, the most important question -Chip-seq- which directly links the TH signaling to gene expression changes, is still unaddressed.

The authors did a "local" scan of motif within 4 kb of ZEB1 promoter regions. However, it is now clear that TF binding is vastly enriched in distal enhancers instead of promoters. It is likely why they could only validate 2 out of 8 genes (ZEB-1, TGF- β 2, N-cadherin, vimentin, Col1a2, 250 Serpine-1, SNAI2 and MAP-1B) in the first place. There are now several new ways to perform chip-seq without good antibodies, including endogenous tagging and cut&tag. Or at least they could try overexpression of a tag protein. In my opinion, a good Chip-seq data will greatly enhance the significance and value of this study. And will provide a great resource for those who are interested in the TH signaling.

Reviewer #4 (Remarks to the Author):

In this revised manuscript, the authors had well addressed all the critiques raised by the reviewers by providing several new data panels and literature review. The manuscript is now suitable for publication in NC.

Reviewer #3 (Remarks to the Author):

I appreciate that the authors have performed extensive revision experiments and have addressed most of my original concerns. However, the most important question -ChIP-seq-which directly links the TH signaling to gene expression changes, is still unaddressed.

We agree with the reviewer that - due the technical difficulties consequent to the lack of functional antibodies- this point has not been fully addressed in the revised manuscript. However, with respect, we disagree with the reviewer on the critical need of ChIP seq in our work. Indeed, we believe that the central question, namely “what is the molecular mechanism by which T3 induces the EMT?” is addressed in our article irrespective of ChIP-seq experiments. Our reasoning is as follows:

We explored the molecular mechanisms by which TH induces EMT by performing multiple and integrated approaches. The EMT profiler analysis indicated a global induction of EMT-related genes by TH signaling. Among the EMT genes induced in the array, ZEB-1 was a strongly up-regulated gene. Given that ZEB-1 is the master inducer of EMP program in cancer (Brabletz, S. 2010; Caramel, J. 2018; Gemmill, R. M. 2011; Spaderna, S. 2008), and given that increased expression of ZEB-1 leads to decreased E-cadherin (epithelial gene) and increased vimentin and N-cadherin expression, we focus our studies on ZEB-1, and found, in ChIP qPCR analysis, that it was indeed a direct, positive target of TH. Importantly, from a functional point of view, ZEB-1 down-regulation was sufficient to drastically reduce the TH-dependent EMT induction. Finally, in vivo analysis of D2KO and D3KO tumors confirmed the modulation of ZEB-1 and E-cadherin in tumors with altered intracellular TH metabolism, thereby supporting the in vitro findings. Overall, these data indicate a molecular mechanism by which TH induces the EMT and invasiveness through ZEB-1.

We believe that, although ChIP seq experiments would help to identify global TH-dependent gene expression changes in SCC tumors, they are not essential to obtain insights into the molecular mechanisms by which TH induces the EMT. Of course, we recognize there are several limitations in ChIP qPCR, and that FOR future studies, ChIP-seq will offer significantly improved data. Integrative analysis of ChIP-seq and epigenetic markers on a genome-wide scale is ongoing in endogenous Tagged-TR α and TR β SCC cell lines in order to understand the genome-wide profiling of TRs proteins and histone modifications.

The authors did a "local" scan of motif within 4 kb of ZEB1 promoter regions. However, it is now clear that TF binding is vastly enriched in distal enhancers in stead of promoters.

We agree with the reviewer and our in silico scan also indicates that multiple TR binding sites are present in the proximal as well as distal promoter gene regions. Thus, by limiting our analysis to the proximal promoter region we missed several relevant regulatory regions. But again, future studies by ChIP seq analysis could provide us more comprehensive data relevant for TH regulation and target gens-

It is likely why they could only validate 2 out of 8 genes (ZEB-1, TGF- β 2, N-cadherin, vimentin, Col1a2, 250 Serpine-1, SNAI2 and MAP-1B) in the first place.

We agree with the reviewer that it is feasible that also many other EMT genes are positive target of T3 and once again (see before).

There are now several new ways to perform chip-seq without good antibodies, including endogenous tagging and cut&tag. Or at least they could try overexpression of a tag protein. In my opinion, a good Chip-seq data will greatly enhance the significance and value of this study. And will provide a great resource for those who are interested in the TH signaling.

We thank the reviewer for his/her suggestions and we plan to perform the knock-in of the Tagged-TR α and TR β in vitro and in vivo mouse models in future studies in order to extend the analysis and obtain a global view on the TH dependent signature relevant in the SCC tumorigenesis.